# Low-Rank Filtering & Smoothing for Sequential Deep Learning

**Joanna Sliwa**  *joanna.sliwa@uni-tuebingen.de*
*Tübingen AI Center, University of Tübingen*

**Frank Schneider**  *f.schneider@uni-tuebingen.de*
*Tübingen AI Center, University of Tübingen*

**Nathanael Bosch**  *nathanael.bosch@uni-tuebingen.de*
*Tübingen AI Center, University of Tübingen*

**Agustinus Kristiadi**  *akristi@uwo.ca*
*Vector Institute, Western University*

**Philipp Hennig**  *philipp.hennig@uni-tuebingen.de*
*Tübingen AI Center, University of Tübingen*

**Reviewed on OpenReview:** *https://openreview.net/forum?id=1TJXpLHLKG*

## Abstract

Learning multiple tasks sequentially requires neural networks to balance retaining knowledge, yet being flexible enough to adapt to new tasks. Regularizing network parameters is a common approach, but it rarely incorporates prior knowledge about task relationships, and limits information flow to future tasks only. We propose a Bayesian framework that treats the network's parameters as the state space of a nonlinear Gaussian model, unlocking two key capabilities: (1) A principled way to encode domain knowledge about task relationships, allowing, e.g., control over which layers should adapt between tasks. (2) A novel application of Bayesian smoothing, allowing task-specific models to also incorporate knowledge from models learned later. This does not require direct access to their data, which is crucial, e.g., for privacy-critical applications. These capabilities rely on efficient filtering and smoothing operations, for which we propose diagonal plus low-rank approximations of the precision matrix in the Laplace approximation (LR-LGF). Empirical results demonstrate the efficiency of LR-LGF and the benefits of the unlocked capabilities.

## 1 Introduction

The central challenge in sequential deep learning is preserving previously acquired knowledge without compromising the ability to learn new information. In *continual learning*, models encounter new tasks one by one—they must acquire each new capability while preserving knowledge from all previous tasks. Despite the model's capacity to learn all tasks *simultaneously*, sequential training often leads to partial "forgetting" of past tasks. Divergence between a sequentially and simultaneously learned model is referred to as *catastrophic forgetting* (McCloskey & Cohen, 1989). Common mitigation strategies constrain the network to encourage changes that maintain prior task performance, e.g. via weight-space regularization (Kirkpatrick et al., 2017). However, an excessive focus on retention can lead to an inability to adapt to new tasks, termed *loss of plasticity* (Dohare et al., 2023). Achieving a good trade-off between learning and forgetting requires a quantification of the model's uncertainty—knowing what parts remain relevant for the previous tasks, and which can be adapted to the new ones. This calls for a principled description of how uncertainty evolves across the sequence of tasks.

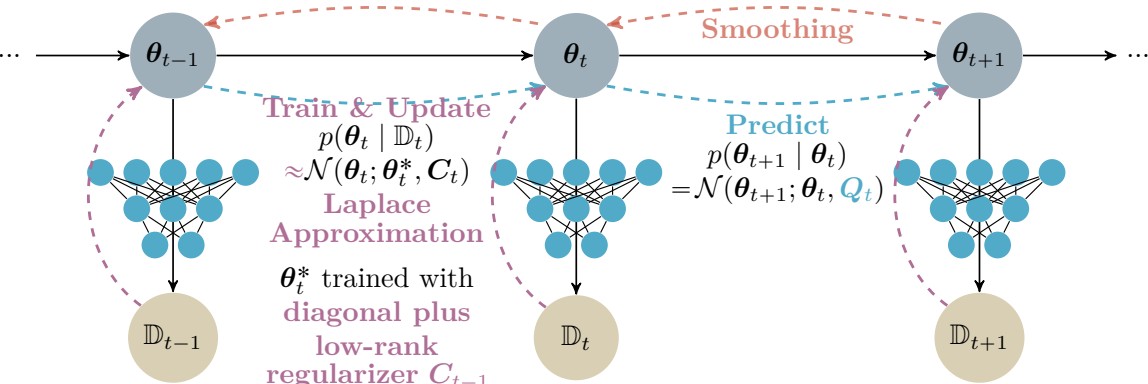

Figure 1: **An efficient weight-space Laplace–Gaussian filter (- - ➤ , - - ➤) and smoother (◄ - -) for sequential deep learning.** We treat the neural network's parameters as a nonlinear Gaussian state-space model and perform efficient inference using diagonal plus low-rank Laplace–Gaussian filtering and smoothing. During the update step (- -➤) we train the neural network on the current task using the parameter covariance as a regularizer, then approximate the posterior distribution with a diagonal plus low-rank Laplace approximation. The predict step (- -➤) adds noise to the model parameters, where the noise covariance $Q$ can be used to model the type of shift between tasks. Smoothing (◄ - -) allows for training task-specific model parameters $\boldsymbol{\theta}_t$ that are informed by *all* tasks, without additional training.

Building on recent work (e.g. Ritter et al., 2018; Chang et al., 2022; 2023), we treat the weight space of the deep network as the state space of a nonlinear Gaussian model. The main advantage of this approach is that it maps the complex problem of sequential learning to the well-understood formalism of Bayesian inference. This enables both efficient computational methods, e.g. Bayesian filtering and smoothing to update beliefs as new tasks arrive and refine them in light of the full sequence, as well as clear conceptual understanding of the state space model's components, e.g. by providing principled ways to include domain knowledge about the sequence of tasks.

**Contributions:** We propose a grounded framework for sequential training of neural networks on related tasks, as in continual, transfer, or online learning, based on a Laplace–Gaussian filter and smoother allowing the following capabilities:

1. **Changes between tasks can be described with established concepts from Bayesian inference.** A Bayesian filter can incorporate domain knowledge about task relationships. Specifically, we study how structured uncertainties can be integrated via the process noise matrix. This matrix models (possible) stochastic drifts between tasks. We demonstrate that, one can integrate prior domain knowledge about how model parameters are likely to change in the *subsequent* related tasks (Section 5.1). For example, it is straightforward to include information indicating that primarily the model's upper (or lower) layers should change between tasks.

2. **Previous tasks can be efficiently informed by knowledge of subsequent tasks.** We examine the benefits of Bayesian smoothing, and sequentially train *task-specific models* that are informed by *all* tasks, without requiring renewed access to the data. We demonstrate that smoothing can significantly boost the performance in initial tasks (Section 5.2). This might be critical in low-data or privacy-critical applications and extends our framework to fields outside of continual learning.

3. **All required operations can be computed efficiently via diagonal plus-low rank approximations.** As part of the filter (Section 3), we employ a weight-space regularizer. Naive filtering/smoothing implementations—even with approximations—would incur prohibitive computational costs. We resolve this by utilizing the Laplace approximation (e.g. MacKay, 1992; Daxberger et al., 2021) with the generalized Gauss-Newton (GGN) matrix (Schraudolph, 2002). By exploiting the low-rank structure of the GGN, we construct *diagonal plus low-rank approximations* of the precision matrices via truncated SVDs. Leveraging

this structure, all required filtering and smoothing operations can be rewritten and computed efficiently (Section 3.2). We provide an open-source JAX implementation of this efficient diagonal plus low-rank Laplace–Gaussian filter, termed LR-LGF[1]. We demonstrate that this diagonal plus-low rank approximation provides a reasonable yet cheap approximation, while enabling efficient filtering and smoothing (Section 5.3).

## 2 Background

**Notation:** We consider supervised learning with a dataset $\mathbb{D} = \{(\boldsymbol{x}_i, \boldsymbol{y}_i) | i = 1, \dots, N\}$ containing training inputs $\boldsymbol{x}_i$ and outputs $\boldsymbol{y}_i$. The objective is to find parameters $\boldsymbol{\theta} \in \mathbb{R}^D$ of a deep network $f_{\boldsymbol{\theta}}$ that minimize a given loss, i.e. $\boldsymbol{\theta}^* = \arg\min_{\boldsymbol{\theta}} \mathcal{L}(\boldsymbol{y}; f_{\boldsymbol{\theta}}(\boldsymbol{x})) = \arg\min_{\boldsymbol{\theta}} \mathcal{L}(\boldsymbol{\theta}, \mathbb{D})$. The first and second derivatives of the loss with respect to the parameters are represented by the gradient $\boldsymbol{g}(\boldsymbol{\theta}) = \nabla_{\boldsymbol{\theta}} \mathcal{L}(\boldsymbol{y}; f_{\boldsymbol{\theta}}(\boldsymbol{x})) \in \mathbb{R}^D$ and the Hessian $\boldsymbol{H}(\boldsymbol{\theta}) = \nabla_{\boldsymbol{\theta}}^2 \mathcal{L}(\boldsymbol{y}; f_{\boldsymbol{\theta}}(\boldsymbol{x})) \in \mathbb{R}^{D \times D}$.

**Continual learning:** While no unified definition exists (Appendix A.1), we describe continual learning as sequentially processing a series of tasks while maintaining good performance on *all* tasks concurrently. Specifically, for a sequence of tasks $t \in \{t_1, \dots, t_T\}$, described by datasets $\mathbb{D}_t$, we prioritize the *average* performance of a single model $f_{\boldsymbol{\theta}}$ across all tasks so far, measured by the average loss (or accuracy) $\frac{1}{T} \sum_t \mathcal{L}(\boldsymbol{\theta}, \mathbb{D}_t)$. Crucially, tasks are experienced sequentially, without access to data from previous tasks. To achieve the required balance between model flexibility and rigidity, we adopt a regularization-based approach (see Section 4), wherein we train the model on task $t$ using a non-isotropic $\ell_2$ regularizer. This regularizer represents prior knowledge and thus constraints from previous tasks (see Kirkpatrick et al., 2017; Ritter et al., 2018). It effectively encodes which weights should remain unchanged to maintain performance on previous tasks and which can be adjusted to perform well on the new task. To implement this, we utilize the Laplace approximation of the Bayesian posterior over the model's parameters, such that the posterior of the previous task becomes the prior for the current one.

**Laplace approximations:** The Bayesian posterior over the model's parameters, $p(\boldsymbol{\theta} \mid \mathbb{D})$, describes the current belief over the specific values of each parameter. It thus reflects (un)certainty about each parameter's value and identifies which parameters still offer flexibility to learn new tasks. The *Laplace approximation* (e.g. MacKay, 1992; Daxberger et al., 2021) constructs a local Gaussian approximation to this typically intractable posterior. It arises from a second-order Taylor expansion of the loss around the maximum a posteriori (MAP) estimate, i.e. the trained $\boldsymbol{\theta}^*$, as $\mathcal{L}(\boldsymbol{\theta}, \mathbb{D}) \approx \mathcal{L}(\boldsymbol{\theta}^*, \mathbb{D}) + \frac{1}{2} (\boldsymbol{\theta} - \boldsymbol{\theta}^*)^\top \boldsymbol{H}(\boldsymbol{\theta}^*) (\boldsymbol{\theta} - \boldsymbol{\theta}^*)$, yielding a Gaussian distribution $p(\boldsymbol{\theta} \mid \mathbb{D}) \approx \mathcal{N}(\boldsymbol{\theta}; \boldsymbol{\theta}^*, \boldsymbol{H}^{-1}(\boldsymbol{\theta}^*))$ known as the Laplace approximation. When used as a weight-space regularizer with strength $\lambda$, this results in the regularized loss for task $t$ as $\mathcal{L}^{\text{reg}}(\boldsymbol{\theta}, \mathbb{D}_t) = \mathcal{L}(\boldsymbol{\theta}, \mathbb{D}_t) + \frac{\lambda}{2} (\boldsymbol{\theta} - \boldsymbol{\theta}_{t-1}^*)^\top \boldsymbol{H}_{t-1}(\boldsymbol{\theta}_{t-1}^*) (\boldsymbol{\theta} - \boldsymbol{\theta}_{t-1}^*)$. Intuitively, we prefer solutions for task $t$ to remain near the previously trained (MAP) parameters $\boldsymbol{\theta}_{t-1}^*$. High-curvature parameters (where changes degrade prior performance) are preserved, while allowing adaptation in low-curvature directions.

**Generalized Gauss-Newton:** Since the Hessian is the second derivative of a composition of two functions, $\mathcal{L}$ and $f$, we can rewrite it with $\boldsymbol{J} \in \mathbb{R}^{D \times C}$ and $\hat{\boldsymbol{H}} \in \mathbb{R}^{C \times C}$ as

$$\boldsymbol{H}(\boldsymbol{\theta}) = \frac{\partial f}{\partial \boldsymbol{\theta}} \frac{\partial^2 \mathcal{L}}{\partial f^2} \frac{\partial f}{\partial \boldsymbol{\theta}}^\top + \frac{\partial^2 f}{\partial \boldsymbol{\theta}^2} \frac{\partial \mathcal{L}}{\partial f} := \boldsymbol{J} \hat{\boldsymbol{H}} \boldsymbol{J}^\top + \frac{\partial^2 f}{\partial \boldsymbol{\theta}^2} \frac{\partial \mathcal{L}}{\partial f} \tag{1}$$

where $C$ denotes the neural network's output dimension. The generalized Gauss-Newton (GGN) matrix is defined as the first term of this expression, $\boldsymbol{J} \hat{\boldsymbol{H}} \boldsymbol{J}^\top$ and it captures loss curvature in output space (Schraudolph, 2002). It is guaranteed to be positive semi-definite for losses convex in the network outputs, making it suitable for covariance estimation. More generally, a GGN can be computed using $M$ mini-batches of size $B$, i.e. $\boldsymbol{J} \hat{\boldsymbol{H}} \boldsymbol{J}^\top = \sum_{m=1}^M \sum_{b=1}^B \boldsymbol{J}_t^{(m,b)} \hat{\boldsymbol{H}}_t^{(m,b)} (\boldsymbol{J}_t^{(m,b)})^\top$, where $\boldsymbol{J}_t^{(m,b)} \in \mathbb{R}^{D \times C}$ is the network's Jacobian with respect to its parameters and $\hat{\boldsymbol{H}}_t^{(m,b)} \in \mathbb{R}^{C \times C}$ is the Hessian of the loss with respect to the neural network outputs, for the $b$-th example of the $m$-th mini-batch. In the next sections, we will use $M = 1$, therefore considering only one mini-batch, which simplifies the computation to: $\boldsymbol{J} \hat{\boldsymbol{H}} \boldsymbol{J}^\top = \sum_{b=1}^B \boldsymbol{J}_t^{(b)} \hat{\boldsymbol{H}}_t^{(b)} (\boldsymbol{J}_t^{(b)})^\top$ whose rank is at most $\min(D, BC)$. For a comparison of the full-batch GGN and full-batch Hessian, as well as, the impact of mini-batching, please see Dangel et al. (2022).

---

[1]Code at: https://github.com/JoannaSliwa/lr-lgf

## 3 A Bayesian Inference Framework for Sequential Learning

We consider a sequence of tasks $t = 1, 2, \ldots, T$ with corresponding datasets $\mathbb{D}_t$. The goal is to compute a posterior distribution over the network's parameters, given the data of all prior tasks, i.e. $p(\boldsymbol{\theta}_t \mid \mathbb{D}_{1:t})$. We are also interested in the posterior distribution for task $t$ given *all* available datasets, i.e. $p(\boldsymbol{\theta}_t \mid \mathbb{D}_{1:T})$. We formulate these distributions as the *filtering* and *smoothing* distributions in a suitable Gaussian state-space model and develop an approximate inference algorithm to efficiently estimate these distributions.

Sequential training can be phrased as a *Bayesian state estimation* problem, where the parameters of the network are treated as the state of a state-space model.

$$\text{Transition model:} \quad p(\boldsymbol{\theta}_{t+1} \mid \boldsymbol{\theta}_t) = \mathcal{N}(\boldsymbol{\theta}_{t+1}; \boldsymbol{\theta}_t, \boldsymbol{Q}), \tag{2a}$$

$$\text{Likelihood:} \quad p(\mathbb{D}_t \mid \boldsymbol{\theta}_t) \propto \exp\left(-\frac{1}{\lambda}\mathcal{L}(\boldsymbol{\theta}_t, \mathbb{D}_t)\right). \tag{2b}$$

The prior transition density $p(\boldsymbol{\theta}_{t+1} \mid \boldsymbol{\theta}_t)$ describes a prior belief over the parameter change from task $t$ to $t+1$, with diagonal Gaussian noise covariance $\boldsymbol{Q} \in \mathbb{R}^{D \times D}$. The un-normalized likelihood $p(\mathbb{D}_t \mid \boldsymbol{\theta}_t)$ encodes the supervised learning task on the dataset $\mathbb{D}_t$, defined by the loss function $\mathcal{L}$, scaled by a factor $1/\lambda \in \mathbb{R}_+$ which controls the regularization strength. This interpretation applies to exponential-family negative-log-likelihood losses. Then, computing the posterior over the weights given the data up to task $t$, that is $p(\boldsymbol{\theta}_t \mid \mathbb{D}_{1:t})$, is known as Bayesian filtering (Särkkä & Svensson, 2023).

In state-space models, the posterior distribution $p(\boldsymbol{\theta}_t \mid \mathbb{D}_{1:t})$ can be computed recursively using the so-called general Bayesian filtering equations (Särkkä & Svensson, 2023):

$$\textbf{Predict step}: \quad p(\boldsymbol{\theta}_t \mid \mathbb{D}_{1:t-1}) = \int p(\boldsymbol{\theta}_t \mid \boldsymbol{\theta}_{t-1}) p(\boldsymbol{\theta}_{t-1} \mid \mathbb{D}_{1:t-1}) \, \mathrm{d}\boldsymbol{\theta}_{t-1}, \tag{3}$$

$$\textbf{Update step}: \quad p(\boldsymbol{\theta}_t \mid \mathbb{D}_{1:t}) \propto p(\mathbb{D}_t \mid \boldsymbol{\theta}_t) p(\boldsymbol{\theta}_t \mid \mathbb{D}_{1:t-1}). \tag{4}$$

These equations demonstrate the value of Bayesian filtering and smoothing for sequential learning as they describe an exact, recursive procedure to learn from a sequence of datasets. However, the exact Bayesian predict and update steps are intractable for all but the simplest state-space models. We will show how to efficiently approximate them with a diagonal plus low-rank Laplace–Gaussian filtering algorithm.

### 3.1 Posterior Approximation: Laplace–Gaussian Filtering

The Laplace–Gaussian filter (LGF) (Koyama et al., 2010) approximates the posterior with Gaussian distributions $p(\boldsymbol{\theta}_t \mid \mathbb{D}_{1:t}) \approx \mathcal{N}(\boldsymbol{\theta}_t; \boldsymbol{m}_t, \boldsymbol{C}_t)$. This approach is commonly known as *Gaussian filtering*, and includes many well-known algorithms such as the extended, and the unscented Kalman filter (Jazwinski, 2007; Särkkä & Svensson, 2023; Julier et al., 2000). The predict and update steps of the LGF are as follows.

**Predict step:**

Since both $p(\boldsymbol{\theta}_{t-1} \mid \mathbb{D}_{1:t-1})$ and $p(\boldsymbol{\theta}_t \mid \boldsymbol{\theta}_{t-1})$ are Gaussian (given LGF assumption and Eq. (2a)), the exact predictive distribution as in Eq. (3) is also Gaussian,

$$p(\boldsymbol{\theta}_t \mid \mathbb{D}_{1:t-1}) = \mathcal{N}(\boldsymbol{\theta}_t; \boldsymbol{m}_t^-, \boldsymbol{C}_t^-), \tag{5}$$

with mean $\boldsymbol{m}_t^- = \boldsymbol{m}_{t-1}$ and covariance $\boldsymbol{C}_t^- = \boldsymbol{C}_{t-1} + \boldsymbol{Q}$. This is exactly equivalent to the predict step of a standard Kalman filter (Kalman, 1960; Kalman & Bucy, 1961).

**Update step:**

The exact update of Eq. (4) is intractable as the likelihood model is not only non-linear, but also non-Gaussian and un-normalized. We therefore apply a Laplace approximation i.e. represent the filtering distribution with a Gaussian distribution

$$p(\boldsymbol{\theta}_t \mid \mathbb{D}_{1:t}) \approx \mathcal{N}(\boldsymbol{\theta}_t; \boldsymbol{m}_t, \boldsymbol{C}_t), \tag{6}$$

where the mean $\boldsymbol{m}_t$ is the mode of the filtering distribution, and $\boldsymbol{C}_t$ is the inverse Hessian. This is known as Laplace–Gaussian filtering (Koyama et al., 2010). More precisely, we take the negative log of the un-normalized posterior, discard constant terms and define the regularized loss $\mathcal{L}_t^{\mathrm{reg}}(\boldsymbol{\theta}_t)$ as

$$\mathcal{L}_t^{\mathrm{reg}}(\boldsymbol{\theta}_t) := \mathcal{L}(\boldsymbol{\theta}_t, \mathbb{D}_t) + \tfrac{1}{2}\big(\boldsymbol{\theta}_t - \boldsymbol{m}_t^-\big)^\top \big(\boldsymbol{C}_t^-\big)^{-1}\big(\boldsymbol{\theta}_t - \boldsymbol{m}_t^-\big) \propto -\log p(\boldsymbol{\theta}_t \mid \mathbb{D}_{1:t}).$$

Then, the mean and covariance of the approximated filtering distribution are given by $\boldsymbol{m}_t = \arg\min_{\boldsymbol{\theta}} \mathcal{L}_t^{\mathrm{reg}}(\boldsymbol{\theta})$ and $\boldsymbol{C}_t = \big(\nabla^2 \mathcal{L}_t^{\mathrm{reg}}(\boldsymbol{m}_t)\big)^{-1}$. The first term is computed via optimization and the second term can be further decomposed into the loss Hessian and the prior covariance $\boldsymbol{C}_t^-$, as

$$\boldsymbol{C}_t = \left(\lambda \nabla^2 \mathcal{L}(\boldsymbol{\theta}, \mathbb{D}_t)\big|_{\boldsymbol{\theta}=\boldsymbol{m}_t} + \big(\boldsymbol{C}_t^-\big)^{-1}\right)^{-1}, \tag{7}$$

which follows from the linearity of the Hessian operator. Following Ritter et al. (2018), we do not include $\lambda$ in the loss function but the covariance update, to prevent the regularization strength from propagating through recursion. In summary, the update step essentially consists of training the network on the new task using a regularized loss function to avoid forgetting previous tasks, and then updating the covariance of the filtering distribution based on the curvature of the un-regularized loss function and the prior covariance.

In dense matrix form, the computational bottlenecks of the above recursion are now explicit. The predict step requires forming the predicted covariance $\boldsymbol{C}_t^- = \boldsymbol{C}_{t-1} + \boldsymbol{Q} \in \mathbb{R}^{D \times D}$, whereas the update step requires computing the task's Hessian and updating it with previous task's covariance, $\boldsymbol{C}_t = \left(\lambda \nabla^2 \mathcal{L}(\boldsymbol{\theta}, \mathbb{D}_t)\big|_{\boldsymbol{\theta}=\boldsymbol{m}_t} + \big(\boldsymbol{C}_t^-\big)^{-1}\right)^{-1}$. Thus, although the Laplace–Gaussian filter has a simple matrix formulation, this remains impractical. We resolve this with low-rank approximations.

## 3.2 Efficient Implementation: Diagonal plus Low Rank

The main bottleneck of the previous algorithm in the context of deep learning lies in computing and storing the dense covariance matrices $\boldsymbol{C}_t, \boldsymbol{C}_t^- \in \mathbb{R}^{D \times D}$ and the exact Hessian. We resolve both issues together by formulating the algorithm to track only diagonal plus low-rank matrices and by using the GGN as a low-rank Hessian approximation, i.e. $\boldsymbol{H}_t \approx \sum_{b=1}^B \boldsymbol{J}_t^{(b)} \hat{\boldsymbol{H}}_t^{(b)} (\boldsymbol{J}_t^{(b)})^\top$.

More precisely, we track diagonal plus low-rank approximations of the *precision* matrices, i.e. the inverse covariance matrices $\boldsymbol{P}_t = \boldsymbol{C}_t^{-1}$, of the form $\boldsymbol{P}_t = \boldsymbol{D}_t + \boldsymbol{U}_t \boldsymbol{\Sigma}_t \boldsymbol{U}_t^\top$, where $\boldsymbol{D}_t \in \mathbb{R}^{D \times D}$ is diagonal, $\boldsymbol{U}_t \in \mathbb{R}^{D \times k}$ is a tall, semi-orthogonal and $\boldsymbol{\Sigma}_t \in \mathbb{R}^{k \times k}$ a dense matrix, with rank $k \ll D$. This resolves the storage and computational issues related to the covariance matrices' size. Storing the diagonal and low-rank matrices requires only $D + Dk + k^2 \ll D^2$ parameters, and the cost of matrix-vector products with the precision matrix is reduced from $\mathcal{O}(D^2)$ to $\mathcal{O}(Dk + k^2)$. What remains is to demonstrate how to preserve the diagonal plus low-rank structure in the predict and update steps.

**Predict step:**

Given a diagonal plus low-rank precision matrix $\boldsymbol{P}_{t-1} = \boldsymbol{D}_{t-1} + \boldsymbol{U}_{t-1} \boldsymbol{\Sigma}_{t-1} \boldsymbol{U}_{t-1}^\top$ and a diagonal process noise covariance $\boldsymbol{Q}$, the predicted precision matrix is also diagonal plus low-rank, with parameters $\big(\boldsymbol{D}_t^-, \boldsymbol{U}_t^-, \boldsymbol{\Sigma}_t^-\big)$ given by

$$\boldsymbol{D}_t^- = \big(\boldsymbol{Q} + \boldsymbol{D}_{t-1}^{-1}\big)^{-1}, \tag{8a}$$

$$\boldsymbol{U}_t^- = \big(\boldsymbol{Q} + \boldsymbol{D}_{t-1}^{-1}\big)^{-1} \boldsymbol{D}_{t-1}^{-1} \boldsymbol{U}_{t-1}, \tag{8b}$$

$$\boldsymbol{\Sigma}_t^- = \left(\boldsymbol{\Sigma}_{t-1}^{-1} + \boldsymbol{U}_{t-1}^\top \boldsymbol{D}_{t-1}^{-1} \boldsymbol{U}_{t-1} - \boldsymbol{U}_{t-1}^\top \boldsymbol{D}_{t-1}^{-\top} \big(\boldsymbol{Q}_{t-1} + \boldsymbol{D}_{t-1}^{-1}\big)^{-1} \boldsymbol{D}_{t-1}^{-1} \boldsymbol{U}_{t-1}\right)^{-1}. \tag{8c}$$

This follows from applying the Woodbury matrix identity twice; full derivation in Appendix C.2.

**Update step:**

As given Eq. (7), the filtering precision matrix $\boldsymbol{P}_t$ is a sum of the predicted precision matrix and the Hessian of the loss function for task $t$. By approximating the full Hessian with the GGN matrix, we can write the

---

**Algorithm 1** Low-rank Laplace–Gaussian Filter (LR-LGF)

---

**Require:** Loss functions $\mathcal{L}_t$ for tasks $t = 1, ..., T$, initial mean $\boldsymbol{m}_0$ and precision $\boldsymbol{P}_0 = \boldsymbol{D}_0 + \boldsymbol{U}_0 \boldsymbol{\Sigma}_0 \boldsymbol{U}_0^\top$, process noise covariance $\boldsymbol{Q}$, regularization strength $\lambda$, rank $k$.
   **for** $t = 1$ **to** $T$ **do**

     **Predict**

1:    $\boldsymbol{m}_t^-, \boldsymbol{D}_t^-, \boldsymbol{U}_t^- \leftarrow \boldsymbol{m}_{t-1}, \left(\boldsymbol{Q} + \boldsymbol{D}_{t-1}^{-1}\right)^{-1}, \boldsymbol{D}_{t-1}^{-1} \boldsymbol{U}_{t-1}$    // *Compute the predicted mean $\boldsymbol{m}_t^-$ and precision $\boldsymbol{P}_t^- = \boldsymbol{D}_t^- + \boldsymbol{U}_t^- \boldsymbol{\Sigma}_t^- \boldsymbol{U}_t^{-\top}$*

     **Update**

2:    $\boldsymbol{\theta}_t^* \leftarrow \arg\min_{\boldsymbol{\theta}} \mathcal{L}_t^{\text{reg}}(\boldsymbol{\theta})$    // *Train on task $t$ with the regularized loss $\mathcal{L}_t^{reg}$*

3:    $\boldsymbol{J}_t^{(b)}, \hat{\boldsymbol{H}}_t^{(b)} \leftarrow (\frac{\partial f}{\partial \boldsymbol{\theta}})^{(b)}, (\frac{\partial^2 \mathcal{L}}{\partial f^2})^{(b)}$   for $b = 1, ..., B$.    // *Compute GGN $\sum_{b=1}^B \boldsymbol{J}_t^{(b)} \hat{\boldsymbol{H}}_t^{(b)} \boldsymbol{J}_t^{(b)\top}$*

4:    $\boldsymbol{D}_t \leftarrow \boldsymbol{J}_t, \hat{\boldsymbol{H}}_t$    // *Approximate the diagonal*

5:    $\tilde{\boldsymbol{U}}_t, \tilde{\boldsymbol{\Sigma}}_t, \tilde{\boldsymbol{V}}_t^\top \leftarrow \text{tSVD}_k(\boldsymbol{W}_t)$    // *Perform a truncated SVD (see Eq. (10)).*

6:    $\boldsymbol{m}_t, \boldsymbol{D}_t, \boldsymbol{U}_t, \boldsymbol{\Sigma}_t \leftarrow \boldsymbol{\theta}_t^*, \boldsymbol{D}_t^-, \tilde{\boldsymbol{U}}_t, \tilde{\boldsymbol{\Sigma}}_t^2$    // *Compute the filtering mean $\boldsymbol{m}_t$ and precision $\boldsymbol{P}_t = \boldsymbol{D}_t + \boldsymbol{U}_t \boldsymbol{\Sigma}_t \boldsymbol{U}_t^\top$*

**Return:** Filtering means $(\boldsymbol{m}_t)_{t=1}^T$ and diagonal plus low-rank precisions $(\boldsymbol{D}_t, \boldsymbol{U}_t, \boldsymbol{\Sigma}_t)_{t=1}^T$.

---

filtering precision matrix as

$$\boldsymbol{P}_t = \boldsymbol{D}_t^- + \boldsymbol{U}_t^- \boldsymbol{\Sigma}_t^- \boldsymbol{U}_t^{-\top} + \sum_{b=1}^B \boldsymbol{J}_t^{(b)} \hat{\boldsymbol{H}}_t^{(b)} \left(\boldsymbol{J}_t^{(b)}\right)^\top. \tag{9}$$

To see that this is a diagonal plus low-rank matrix, with increased rank, we denote matrix square-roots by $\boldsymbol{A}^{1/2}$, with $\boldsymbol{A}^{1/2}(\boldsymbol{A}^{1/2})^\top = \boldsymbol{A}$, and define $\boldsymbol{W}_t \in \mathbb{R}^{D \times (k + BC)}$ as

$$\boldsymbol{W}_t := \left[\boldsymbol{U}_t^- \left(\boldsymbol{\Sigma}_t^-\right)^{1/2} \quad \boldsymbol{J}_t^{(1)} \left(\hat{\boldsymbol{H}}_t^{(1)}\right)^{1/2} \quad \cdots \quad \boldsymbol{J}_t^{(B)} \left(\hat{\boldsymbol{H}}_t^{(B)}\right)^{1/2}\right]. \tag{10}$$

The filtering precision matrix can then be written $\boldsymbol{P}_t = \boldsymbol{D}_t^- + \boldsymbol{W}_t \boldsymbol{W}_t^\top$, but with increased rank $k + BC$. To prevent rank inflation, we compress the matrix $\boldsymbol{W}_t$ by performing a truncated singular value decomposition (SVD) of rank $k$ to obtain $\boldsymbol{W}_t \approx \tilde{\boldsymbol{U}}_t \tilde{\boldsymbol{\Sigma}}_t \tilde{\boldsymbol{V}}_t^\top$, with $\tilde{\boldsymbol{U}}_t \in \mathbb{R}^{D \times k}$, $\tilde{\boldsymbol{V}}_t \in \mathbb{R}^{k \times D}$, and diagonal $\tilde{\boldsymbol{\Sigma}}_t \in \mathbb{R}^{k \times k}$. Additionally, we approximate the diagonal $\boldsymbol{D}_t$ before the tSVD as explained in Appendix C.4. Altogether, the filtering precision matrix is

$$\boldsymbol{P}_t = \boldsymbol{D}_t + \tilde{\boldsymbol{U}}_t \tilde{\boldsymbol{\Sigma}}_t \tilde{\boldsymbol{V}}_t^\top \tilde{\boldsymbol{V}}_t \tilde{\boldsymbol{\Sigma}}_t \tilde{\boldsymbol{U}}_t^\top = \boldsymbol{D}_t + \tilde{\boldsymbol{U}}_t \tilde{\boldsymbol{\Sigma}}_t^2 \tilde{\boldsymbol{U}}_t^\top =: \boldsymbol{D}_t + \boldsymbol{U}_t \boldsymbol{\Sigma}_t \boldsymbol{U}_t^\top. \tag{11}$$

The resulting diagonal plus low-rank approximation of the filtering precision matrix enables the Laplace–Gaussian filter (Section 3.1) for continual deep learning (Algorithm 1).

### 3.3 Task-Specific Models via Backwards Smoothing

Until now, our focus has been on computing and storing a single model trained sequentially on tasks. However, if the tasks differ, it may be beneficial to store *task-specific* models, that are informed by all datasets—still with the restriction that the datasets are only observed sequentially. For state-space models, this is known as *smoothing* (Rauch et al., 1965; Särkkä & Svensson, 2023). Since we consider Gaussian filtering distributions, and the transition model is linear and Gaussian (Eq. (2a)), the smoothing distribution is also Gaussian, i.e. $p(\boldsymbol{\theta}_t \mid \mathbb{D}_{1:T}) \approx \mathcal{N}(\boldsymbol{\theta}_t; \boldsymbol{m}_t^s, \boldsymbol{C}_t^s)$, and its mean and covariance can be computed recursively backwards in time. That is, after first performing the forward filtering pass over $t = 1, ..., T$, we obtain the smoothed distributions by propagating information from later tasks back to earlier ones. Starting with $\boldsymbol{m}_T^s = \boldsymbol{m}_T$ and $\boldsymbol{C}_T^s = \boldsymbol{C}_T$, the smoothing equations are given by (Rauch et al., 1965)

$$\boldsymbol{m}_t^s = \boldsymbol{m}_t + \boldsymbol{G}_t \left(\boldsymbol{m}_{t+1}^s - \boldsymbol{m}_{t+1}^-\right), \tag{12a}$$

$$\boldsymbol{C}_t^s = \boldsymbol{C}_t + \boldsymbol{G}_t \left(\boldsymbol{C}_{t+1}^s - \boldsymbol{C}_{t+1}^-\right) \boldsymbol{G}_t^\top, \tag{12b}$$

where $\boldsymbol{G}_t = \boldsymbol{C}_t \left(\boldsymbol{C}_{t+1}^-\right)^{-1}$ is known as the *smoothing gain*. The smoothing equations can be formulated in terms of precision matrices, and if the filtering precision matrix is diagonal plus low-rank, then the smoothing precision matrix is also diagonal plus low-rank (full derivation in Appendix C.4) where the smoothing gain takes form $\boldsymbol{G}_t = \boldsymbol{D}_t^G + \boldsymbol{U}_t^G \boldsymbol{\Sigma}_t^G (\boldsymbol{V}_t^G)^\top$, with

$$\boldsymbol{D}_t^G := \left(I + \boldsymbol{Q}\boldsymbol{D}_t\right)^{-1}, \tag{13}$$

$$\boldsymbol{U}_t^G := \left(I + \boldsymbol{Q}\boldsymbol{D}_t\right)^{-1} \boldsymbol{Q}U_t, \tag{14}$$

$$\boldsymbol{\Sigma}_t^G := -\left(\boldsymbol{\Sigma}_t^{-1} + \boldsymbol{U}_t^\top \left(I + \boldsymbol{Q}\boldsymbol{D}_t\right)^{-1} \boldsymbol{Q}U_t\right)^{-1}, \tag{15}$$

$$\left(\boldsymbol{V}_t^G\right)^\top := \boldsymbol{U}_t^\top \left(I + \boldsymbol{Q}\boldsymbol{D}_t\right)^{-1}. \tag{16}$$

Eq. (12) also shows why the diagonal plus low-rank structure is crucial for efficient computation of the smoothing means. If $\boldsymbol{G}_t \in \mathbb{R}^{D \times D}$ were dense, the matrix-vector product would have a prohibitive computational cost of $\mathcal{O}(D^2)$, but using $\boldsymbol{G}_t = (\boldsymbol{P}_t)^{-1} \boldsymbol{P}_{t+1}^-$ and implementing the matrix vector product as a sequential product with two diagonal plus low-rank matrices, the computational cost is reduced to $\mathcal{O}(Dk + k^2)$ and smoothing becomes feasible.

## 4 Related Work

**Continual learning:** Wang et al. (2024) categorizes approaches to continual learning as either *regularization-*, *optimization-*, *representation-*, *architecture-*, or *replay*-based (Appendix A.2). In this paper, we use a (weight) regularization-based approach which tackles catastrophic forgetting in neural networks by constraining the model's weight space. This approach aims to identify weights that are important for the previous tasks, and constrains their changes in subsequent tasks. A scalar hyperparameter $\lambda$, used on the regularizer, allows trading off performance on previous tasks vs. the ability to learn new tasks. *Elastic Weight Consolidation* (EWC) (Kirkpatrick et al., 2017) takes inspiration from neuroscience and places a quadratic constraint on the model's parameters through a regularized loss. In each task, the regularizer consists of the sum of penalties of previous tasks where the importance of the weights is captured by the diagonal Fisher information matrix. In contrast, *Online Structured Laplace Approximations* (OSLA) (Ritter et al., 2018) uses a block-diagonal K-FAC (Martens & Grosse, 2015) Hessian approximation. Their regularizer consists of a single penalty, recursively updated with the most recent log likelihood scaled by $\lambda$. The authors observe improved performance over EWC, attributed to a more expressive Hessian approximation also capturing parameter interactions within a layer. For the exact definitions, see Appendix A.2. Other weight regularization-based methods try to find better ways to represent the importance measure (Zenke et al., 2017), refine the penalty (Liu et al., 2018; Park et al., 2019), use a expansion-renormalization approach (Lee et al., 2017; Schwarz et al., 2018), or target the network (Nguyen et al., 2018). EWC and OSLA are closest to our approach since we update the precision of our penalty and scale the regularizer similarly. However, they use different curvature approximations and neither method uses a Bayesian filter or smoother. For related work on Gaussian Processes for continual learning, see Appendix A.2.

**Bayesian filtering and smoothing:** A well-established formalism for sequential data is Bayesian filtering and smoothing (Särkkä & Svensson, 2023). Learning neural network weights via an extended Kalman filter (EKF) was proposed already in the '90s (Singhal & Wu, 1988; Feldkamp et al., 1998), but the EKF and related methods (Jazwinski, 2007; Särkkä & Svensson, 2023; Julier et al., 2000) suffer from quadratic memory and cubic computational costs, which is prohibitive for deep learning. Therefore, various approximations have been proposed. Diagonal EKF variants for both training and online learning of neural networks exist, but as they ignore interactions between the weights their quality is often limited (Puskorius & Feldkamp, 1991; Chang et al., 2022). Treating only the network's last layer probabilistically makes the Kalman filter tractable, but also reduces expressiveness (Titsias et al., 2024). Recently, diagonal plus low-rank approximations have been proposed as a more expressive alternative and the resulting EKF variant, called LO-FI, has been shown to be effective for online learning from streaming data (Chang et al., 2023). Building on LO-FI, we focus on continual learning, where each task's data $\mathbb{D}_t$ is observed in its entirety and performance across past tasks matters. Therefore, instead of the EKF we use a Laplace–Gaussian filter (Koyama et al., 2010), compute the MAP estimate in each filtering step via optimization, and compute the posterior with a diagonal plus

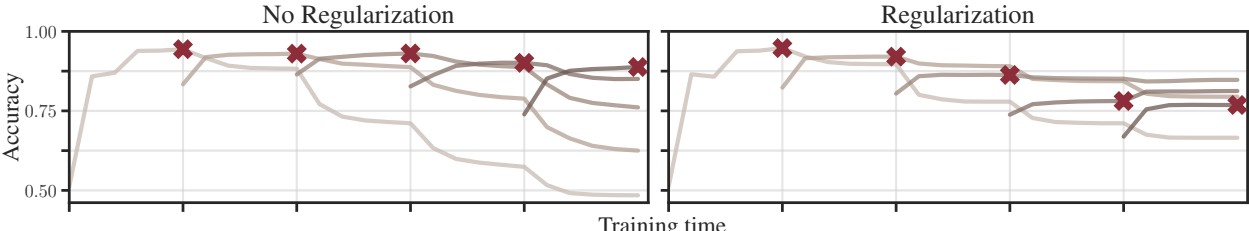

Figure 2: **Regularization helps with forgetting but affects flexibility.** Epoch-wise evaluation accuracy during training on GRADUAL CAMELYON. (*Left*) without regularization; (*right*) with regularization. Each colored curve shows the evaluation accuracy for one task over the course of training, ordered from task 1 (—) through tasks 2, 3, and 4 (— — —) to the final task 5 (—). Red crosses (✖) indicate each task's final accuracy at the end of its own training phase. Regularization mitigates forgetting of the past tasks, although it impacts plasticity when learning new tasks.

low-rank Laplace approximation. This differs strongly from the EKF and is more akin to the more accurate *iterated* EKF (IEKF) (Bell & Cathey, 1993; Särkkä & Svensson, 2023). Additionally, we propose the use of a smoother to improve performance on past tasks for task-specific models. Kalman filtering has recently been explored for fine-tuning large models. LoKO (Abdi et al., 2025) applies a Kalman filter for online fine-tuning, leveraging LoRA and a diagonal covariance approximation to improve convergence in vision and language tasks. Another application of Bayesian filtering has been explored by Lee & Chang (2024) for online test-time adaptation, addressing domain adaptation under varying distribution shifts.

**Low-Rank plus Diagonal:** Chang et al. (2023) introduce an efficient implementation of the predict and update steps, where the authors obtain a diagonal plus low rank via the Woodbury identity. Then, they update the precision with GGN and project to a smaller rank via SVD. Our approach follows the same fundamental steps. Our contribution is to apply this framework to sequential learning with task priors via $Q$ and backward smoothing, while the diagonal plus low rank form makes the approach feasible for neural networks. The diagonal plus low-rank approximation of the covariance matrix has been explored for variational inference in large-scale models (Mishkin et al., 2019; Lambert et al., 2023).

## 5 Experiments

We showcase the benefits of the filtering framework by first studying how domain knowledge can be integrated via the dynamics model (Section 5.1). Next, we examine the benefits of smoothing and find that it can boost the performance of task-specific models learned "earlier", without renewed access to data (Section 5.2). Lastly, we analyze our diagonal plus low-rank approximation of the GGN, used to make the filtering and smoothing operations efficient, comparing it to other weight-space regularizers proposed in the literature (Section 5.3).

### 5.1 Integrating Domain Knowledge via Q

We examine the benefits of incorporating domain knowledge, i.e. how tasks are related, via the dynamics model, specifically, the process noise matrix $Q$. Roughly speaking, $Q$ adds uncertainty to the next task's model parameters (Eq. (2a)), describes task relationship, and thus controls the model's flexibility. For $Q = 0$, we assume all sub-tasks $t$ belonging to a shared meta-task, or that the optimal model parameters do not change between tasks. In contrast, by using a structured $Q$, we can incorporate how we believe the tasks, and therefore the model parameters, change, e.g. indicating that mostly the lower layers of the network change. This could be useful in Bayesian optimization, for example, in material discovery with the use of foundation models (Kristiadi et al., 2024).

**Setting:** We empirically examine this on the CAMELYON dataset (Koh et al., 2021), which consists of images of either healthy or cancerous cells collected from different hospitals. To showcase the benefits of $Q$ (and the smoother in Section 5.2), we create a continual learning task with an *ordered* series of tasks that we

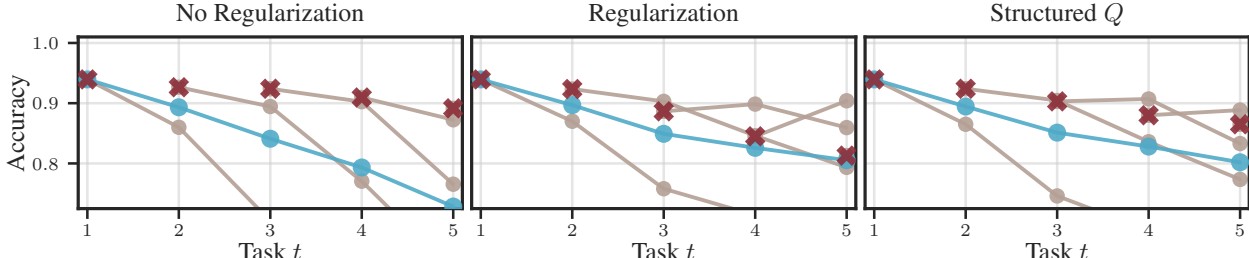

Figure 3: **The effect of $Q$ on the average and current task's performance.** *(Left)* Without regularization, we see a significant drop in the average performance across all seen tasks (—), while the performance on the current task (✖) is strong. *(Center)* Adding regularization, helps boost the average performance across tasks, but to the detriment of the current task (most notably task $t=5$). *(Right)* Using a structured $Q$ can boost the current task performance, while keeping the same average performance across tasks. See Fig. 12 in Appendix D.4 for a summary of the same experimental results across 8 random seeds.

term GRADUAL CAMELYON, by gradually changing the brightness of the samples between tasks. Task $t=1$ has the darkest and task $t=5$ has the brightest pixels. We use a network with three convolutional and two dense layers (Appendix D.2 provides full details).

**Results:** To introduce the problem setting, Figure 2 shows the forgetting of earlier tasks across training when no regularization is used, and how regularization can mitigate this forgetting. We notice that while regularization helps boost the model's average performance across tasks, it comes at the detriment of the current task's performance. In Figure 3, we re-introduce flexibility by adding a non-zero process noise $Q$ (Fig. 3 *(right)*) to this regularization. This prioritizes the current task (see also Fig. 12 *(right)*) while maintaining the same average accuracy (Fig. 12 *(center)*). Across experiments, a structured transition noise (i.e. non-zero values on the lowest layers) performs slightly better than a simpler isotropic, scalar-times-identity $Q$ (Fig. 12). The structured $Q$ encodes an intuition that brightness changes likely affect the lower convolutional layers the most. It demonstrates how structural domain knowledge about task relationship can be explicitly included in the filtering framework, in this case to prioritize local performance.

A non-zero $Q$ indicates that tasks—and thus optimal model parameters—differ. In those cases, it may be beneficial to store *task-specific models* that still share information across tasks instead of a single model for all tasks, which we will explore, via smoothing, next.

## 5.2 Going Back in Time via Smoothing

We investigate the benefits of using a smoother to imbue previous tasks with the knowledge gained from later tasks. We use the experimental setting of Section 5.1 with fewer data points per task. Crucially, we now learn task-specific models $\boldsymbol{\theta}_t$, for each task $t=1,\ldots,5$. As the filter computes posteriors $p(\boldsymbol{\theta}_t \mid \mathbb{D}_{1:t})$, the model for the last task $\boldsymbol{\theta}_5$ has incorporated information from all previous tasks. However, the model for the first task $\boldsymbol{\theta}_1$ is still only informed by the first task. Using a Bayesian smoother (Section 3.3), we can also update the first model with subsequent knowledge, *without any additional access to any data*. We simply update earlier model parameters based on the later model parameters. This is relevant for privacy-sensitive settings, by maintaining separate models for each task, making it ideal when data or models cannot be shared. For example, for electronic health records, where hospitals face strict data-sharing restrictions or retention policies (Moreno-Muñoz et al., 2019). On-device learning is another key application, with domains such as domestic robots or monitoring cameras (Kapoor et al., 2021; Verwimp et al., 2024).

**Results:** Figure 4 illustrates the benefits of smoothing on GRADUAL CAMELYON. Smoothing passes information backwards, informing earlier models about the knowledge gained on later tasks, increasing performance (Fig. 4 *(left)*). Across all 8 seeds and tasks, we see a performance increase when comparing the filtered model to the smoothed model (Fig. 4 *(right)*). In particular, accuracy for the second task increases substantially from 70.5% to 75.9%. Smoothing may be particularly useful in sequences of tasks with only little

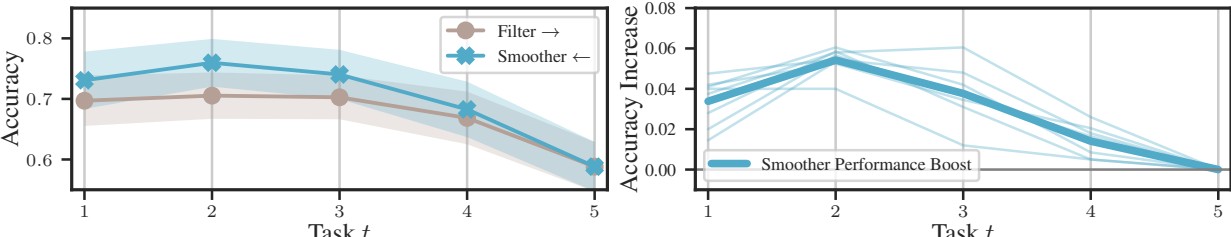

Figure 4: **Smoothing improves the performance on earlier tasks.** *(Left)* Task-wise performance after filtering (—) or smoothing (—) up to task $t$, with shaded regions representing one standard deviation across 8 seeds. *(Right)* The smoother consistently improves performance by incorporating information from all tasks without accessing their data. The thick line is the mean improvement of smoothed vs. the filtered model with the thin lines showing the improvements for each seed.

data—and thus transferring knowledge between tasks is required—and the tasks are substantially different or challenging that a single model cannot accurately learn all tasks.

Additionally, smoothing can also provide a grounded way to infer model parameters for novel tasks between two observed tasks. Inferring a neural network's trained parameters for tasks for which we never directly observed any data, e.g. for intermediate brightness levels, provides an interesting avenue for future work.

### 5.3 Diagonal Plus Low-Rank Approximations

Lastly, we analyze our proposed diagonal plus low-rank curvature approximation, which allows efficient computation of all filtering and smoothing operations. Here, we evaluate it outside of the filtering and smoothing context, isolating the effectiveness of this approximation. We compare it to common approximations from the continual learning literature. The results show that adding a low-rank part—even with very small rank—adds useful information, while still offering the advantage of efficient filtering and smoothing.

**Setting:** We compare our diagonal plus low-rank approximation of the GGN to both EWC (where the regularizer differs as shown in Appendix A.2 and uses diagonal approximations of the Fisher matrix) and OSLA (which is the most similar but uses block-diagonal K-FAC approximations of the Hessian). The comparison is performed on standard continual learning benchmarks. In *Permuted MNIST*, each task $t \in \{1, \ldots, 5\}$ consist of classifying the MNIST digits, where each task uses a random (but for this task fixed) permutation of the image pixels. The *Disjoint MNIST* setting consists of 2 tasks: task $t = 1$ includes images with labels $y \in \{1, \ldots, 4\}$, while $t = 2$ contains labels $y \in \{5, \ldots, 9\}$. For both continual learning problems, we train a small 2-layer MLP with 400 hidden units per layer sequentially on all tasks (see Appendix D.2 for full details). Additionally, we evaluate LR-LGF on two more realistic continual learning problems, based on the CAMELYON and IWILDCAM datasets from the WILDS benchmark (Koh et al., 2021). We train on GRADUAL CAMELYON, described in Section 5.1, and on LOCATIONSPLIT IWILDCAM, where each task consists of classifying the species in images from disjoint location (Appendix D.1). We use a network with three convolutional and two dense layers (Appendix D.2 provides full details). After each task, we record the current model's performance on all tasks seen so far. For each problem and method, we tune the regularization strength $\lambda$ on a grid, maximizing the average accuracy. Each experiment is repeated with 8 random seeds. In these experiments, we deliberately disable the filter and smoother (i.e. $Q = 0$) to focus exclusively on the quality of the curvature approximation for continual learning. This limits the experiments to the shared functionality of EWC, OSLA, and our method, intentionally disregarding the additional benefits of our approach. Lastly, we train a model on all tasks jointly as an upper bound in a given setting.

**Results:** Table 1 and Fig. 8 show that regularizing with a diagonal plus low-rank GGN approximation consistently reduces forgetting in all settings compared to no regularization. Additionally, our method outperforms EWC, indicating that the low-rank component captures useful uncertainty structure beyond EWC's purely diagonal approximation. Our method is also competitive with OSLA across all settings, achieving the strongest overall results by matching OSLA on LOCATIONSPLIT IWILDCAM and outperforming

Table 1: **Comparing our diagonal low-rank GGN approximation to other regularization methods**. We report the final *average* accuracy ($\pm$ one standard deviation across 8 seeds) on all sub-tasks after sequentially learning each task.

| | *Permuted* MNIST | *Disjoint* MNIST | GRADUAL CAMELYON | LOCATIONSPLIT iWILDCAM |
|---|---|---|---|---|
| **LR-LGF** ($Q\!=\!0$) | $0.967_{\pm0.004}$ | $0.692_{\pm0.019}$ | $0.772_{\pm0.036}$ | $0.504_{\pm0.021}$ |
| EWC | $0.931_{\pm0.024}$ | $0.676_{\pm0.029}$ | $0.733_{\pm0.052}$ | $0.497_{\pm0.006}$ |
| OSLA | $0.953_{\pm0.009}$ | $0.682_{\pm0.044}$ | $0.725_{\pm0.032}$ | $0.504_{\pm0.008}$ |
| Baseline | $0.664_{\pm0.020}$ | $0.460_{\pm0.005}$ | $0.727_{\pm0.023}$ | $0.492_{\pm0.005}$ |
| Joint | $0.978_{\pm0.002}$ | $0.979_{\pm0.001}$ | $0.952_{\pm0.002}$ | $0.588_{\pm0.020}$ |

Figure 5: **The low-rank approximation across training.** *(Top)* Our rank $k\!=\!10$ approximation to the precision matrix across tasks on *Permuted* MNIST (see Fig. 9 for $k=20$). *(Bottom)* Histograms of the eigenvalues of the approximation's low-rank part. With growing $t$, the eigenvalues increase in magnitude, indicating larger certainty and less flexibility.

it on the remaining benchmarks. This suggests that rich curvature structures are beneficial—extending our framework to support efficient filtering and smoothing with block-diagonal K-FAC approximations is therefore a promising direction for future work.

**Properties of the diagonal plus low-rank approximation:** We further analyze the behavior of our diagonal plus low-rank approximation during training. Figure 5 visualizes its evolving eigenvalues after training each task. For $Q\!=\!0$, the diagonal part remains constant throughout training (Eq. (8)). First, the diagonal plus low-rank approximation is significantly more expressive than diagonal ones like EWC, with the low-rank component capturing useful parameter relationships. Second, relatively low ranks and Hessian batch size are sufficient to achieve a meaningful yet cheap approximation. Increasing the rank beyond $\approx 10$ (the number of classes) or the batch size beyond $\approx 8$ offers little additional benefit (Figs. 9 to 11). Lastly, the magnitudes of the eigenvalues grow as training progresses, reflecting increased certainty in parameter values. Since the top eigenvalues are computed as part of the truncated SVD (line 7 in Algorithm 1), they can serve as a cost-effective diagnostic. For instance, they could indicate whether the network's "memory" has reached capacity, and the ratio of retained to deflated eigenvalues may inform the choice of the rank $k$.

## 6 Conclusion

**Limitations:** Our diagonal plus low-rank GGN approximation stores matrices of size $1\times D$, $D\times C$, $C\times C$. Thus, LR-LGF is beneficial if $C\ll D$, as in our image-classification benchmarks, but imposes constraints if the number of classes $C$ is large. For very large output spaces, one may choose last-layer, block-diagonal, layer-wise, or K-FAC-style approximations. However, these reduce expressivity, and efficient filtering and smoothing is not straightforward. Additionally, low-rank approximations with truncated SVDs can lead to

an overestimated curvature, which can hurt performance. While incorporating task-specific knowledge into the dynamics model is desirable, it can be challenging to set the associated parameters well in practical applications. Finally, our experiments were designed to showcase specific scenarios serving as proxies for real-world continual learning challenges and to highlight our method's strengths. The experiments presenting the benefits of $Q$ and the backward smoother should be viewed as proof-of-concept demonstrations rather than broad real-world validation.

**Summary and discussion:** We have presented an efficient low-rank Laplace–Gaussian filtering framework for sequential deep learning across multiple related tasks. The method treats the network's weights as states, and the individual tasks as likelihood models in a Bayesian state-space model. A diagonal plus low-rank Gaussian approximation allows for efficient approximate inference when combined with a low-rank Laplace approximation via the GGN matrix to compute the filtering distributions, i.e. the posterior over the network's weights given all past and present tasks. We leverage this formalism to compute *task-specific* models via Bayesian smoothing, incorporating knowledge from later tasks into earlier models *without renewed access to the data*. Our methodology maps the complex problems of sequential or continual learning to well-understood Bayesian filtering and smoothing. This addresses two key challenges of continual learning from Verwimp et al. (2024): It raises computational efficiency, and conceptually clarifies the method's components, casting the relations between datasets through the observation and dynamics models of Markov Chains.

## Acknowledgments

The authors gratefully acknowledge co-funding by the Carl Zeiss Foundation, (project "Certification and Foundations of Safe Machine Learning Systems in Healthcare") and the European Union (ERC, ANUBIS, 101123955). Views and opinions expressed are however those of the author(s) only and do not necessarily reflect those of the European Union or the European Research Council. Neither the European Union nor the granting authority can be held responsible for them. Philipp Hennig is a member of the Machine Learning Cluster of Excellence, funded by the Deutsche Forschungsgemeinschaft (DFG, German Research Foundation) under Germany's Excellence Strategy – EXC number 2064/1 – Project number 390727645; he also gratefully acknowledges the German Federal Ministry of Education and Research (BMBF) through the Tübingen AI Center (FKZ: 01IS18039A); and funds from the Ministry of Science, Research and Arts of the State of Baden-Württemberg. Frank Schneider is supported by funds from the Cyber Valley Research Fund. Joanna Sliwa and Nathanael Bosch are grateful to the International Max Planck Research School for Intelligent Systems (IMPRS-IS) for support. Further, we are grateful to Marvin Pförtner for the helpful discussions and the members of Methods of Machine Learning group for providing feedback to the manuscript.

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

# A Continual Learning

## A.1 Definition

To clarify our notion of continual learning, we draw on several representative definitions from the literature. Continual learning is generally understood as the ability to learn tasks sequentially while preserving knowledge from previously encountered tasks, even when their data are no longer available (Kirkpatrick et al., 2017). The broader aim is to build models that perform well across multiple tasks and can incorporate new information over time without extensive retraining on stored past data (Ritter et al., 2018; Titsias et al., 2020b) This is commonly formalized as minimizing the total loss over all tasks while having access only to data from the current task during training (Zenke et al., 2017). Recent review on continual learning (Wang et al., 2024) emphasizes that the goal is to behave as if all tasks had been observed jointly, which requires balancing learning flexibility, memory stability, and generalization across tasks. Mirzadeh et al. (2021) point out that when the model has access to all the data, it learns different solutions than when learning continuously. The book of lifelong learning (Chen & Liu, 2018) thoroughly defines continual learning and its connections to transfer, multi-task, online and reinforcement learning. The authors define that at a given point in time, a model has learned a sequence of previous tasks $t = 1, 2, \dots, T$ with corresponding data $\mathbb{D} = \{\mathbb{D}_1, \mathbb{D}_2, \dots, \mathbb{D}_T\}$. The tasks can be of the same or different type/domain. When the model encounters a new task $t_{T+1}$ with data $\mathbb{D}_{T+1}$ it can leverage the past knowledge and learn the new task. Then it implements the new knowledge into the existing knowledge base. The main requirements are that the learning is in a continuous fashion, the knowledge is accumulated and the knowledge from the previous tasks can be used to learn new ones.

In this work, we have tested our method for domain-incremental learning, where the same classes appear in different contexts, as well as, task-free continual learning, for disjoint label spaces with no task identity provided.

## A.2 Additional Approaches to Continual Learning

In Section 4, we focus on regularization-based approaches to continual learning, as this is the approach used in the paper. Below, we present the summary of the used competing methods:

**EWC:** Based on the descriptions in the original paper (Kirkpatrick et al., 2017), as well as, mentions in subsequent papers (Aich, 2021; van de Ven & Tolias, 2019) we use the following loss function when learning task $t$:

$$\mathcal{L}_t^{\text{reg}}(\theta) = \mathcal{L}(\boldsymbol{\theta}) + \frac{\lambda}{2} \sum_{k}^{t-1} (\boldsymbol{\theta}_t - \boldsymbol{\theta}_k^*)^T \boldsymbol{\Lambda}_k (\boldsymbol{\theta}_t - \boldsymbol{\theta}_k^*)$$

$T$ is the number of all tasks, $\boldsymbol{\theta}^*$ is the mode for given task, $\boldsymbol{\Lambda}_k$ is the diagonal of the Fisher Information Matrix of the negative log-likelihood $F$, given by

$$F = \frac{1}{|\mathcal{S}_k|} \sum_{\boldsymbol{s}_k}^{\mathcal{S}_k} \left( -\frac{\delta \log p(\boldsymbol{s}_k|\boldsymbol{\theta})}{\delta \boldsymbol{\theta}} \bigg|_{\boldsymbol{\theta} = \boldsymbol{\theta}_k^*} \right)^2$$

where $\mathcal{S}_k$ is a subset of the data.

**OSLA:** We use the following loss funtion when learning task $t$ based on the original paper (Ritter et al., 2018):

$$\mathcal{L}_t^{\text{reg}}(\theta) = \mathcal{L}(\boldsymbol{\theta}) + \frac{1}{2} (\boldsymbol{\theta}_t - \boldsymbol{\theta}_{t-1}^*)^T \boldsymbol{\Lambda}_{t-1} (\boldsymbol{\theta}_t - \boldsymbol{\theta}_{t-1}^*)$$

where $\boldsymbol{\Lambda}_{t-1}$ is sum of the scaled Hessian of the negative log-likelihoods:

$$\boldsymbol{\Lambda}_{t-1} = \lambda \left( -\frac{\delta^2 \log p(\mathbb{D}_{t-1}|\boldsymbol{\theta})}{\delta \boldsymbol{\theta} \delta \boldsymbol{\theta}} \bigg|_{\boldsymbol{\theta} = \boldsymbol{\theta}_{t-1}^*} \right) + \boldsymbol{\Lambda}_{t-2}$$

The Hessian is approximated as a block-diagonal Kronecker factorization $\boldsymbol{A}_{i-1} \otimes \boldsymbol{G}_i$ where $\boldsymbol{A}_{i-1}$ is the covariance of the input to layer $i$ and $\boldsymbol{G}_i$ is the covariance of the output of layer $i$.

For completeness, we briefly note that outside of regularization-based continual learning methods, other approaches are commonly categorized into replay-based, optimization-based, representation-based, and architecture-based approaches; we refer the reader to Wang et al. (2024) for a comprehensive review.

**Gaussian Processes for CL:** The state-space model that we consider in this work has a continuous-time representation as a Gaussian process with a Wiener process kernel. The GP maps the task-id to the neural network weights, that is $(t \mapsto \boldsymbol{\theta}_t)$, but the actual model of interest which maps data inputs to outputs, i.e. $(\boldsymbol{x} \mapsto \boldsymbol{y})$, is the neural network inside the observation model. In the related works (Moreno-Muñoz et al., 2019; Kapoor et al., 2021), the GP is used to learn the function of interest $(\boldsymbol{x} \mapsto \boldsymbol{y})$. There is no fundamental distinction between tasks and thus there is no prior which encodes how continuous or distinct weights between those should be, which is precisely the role of the GP in our work. Titsias et al. (2020a) combines GPs and NNs and considers GPs with deep kernel functions, but here again the GP is an integral part of the function of interest as it outputs the quantity of interest $\boldsymbol{y}$. In contrast, the GP in our work plays a much more subtle role and only provides a prior over the neural network weights varying per task, and it is the neural network that plays the more prominent role and maps the data inputs to outputs.

**Sharpness-aware minimization :** Sharpness-aware minimization (SAM) (Foret et al., 2021) seeks parameters whose neighborhoods have low loss, typically through a minimax objective over parameter perturbations. Related variants (Dauphin et al., 2024; Singh et al., 2025), further study which curvature-related terms drive sharpness regularization. In contrast, our regularizer uses curvature accumulated from previous tasks: it penalizes movement in directions important for earlier tasks, as measured by the previous task's GGN. Thus, SAM-style methods primarily target generalization through sharpness control, whereas our method targets the stability–plasticity trade-off in sequential learning.

**Task similarity:** The similarity of tasks in continual learning was studied in previous works (Adel, 2024; Ramesh & Chaudhari, 2022; Wang et al., 2022). Importantly, the authors state that one may benefit from the knowledge of task relatedness. In such a setting, we can employ an ensemble of small models that grows when competing (vs synergistic) tasks appear. When dissimilar tasks are learnt by one model, the tasks compete for the fixed capacity of the model. Additionally, evaluating the task similarity and initializing some components with the most similar past task' one may help with the training.

## B    Implementation Details

---

**\_\_\_init\_\_\_**

Creates a diagonal matrix scaled by $\lambda_{\text{INIT}}$ and a zero low-rank term.

**Require:** dimension $D$, scale $\lambda_{\text{INIT}}$
1: $\boldsymbol{D} \leftarrow \lambda_{\text{INIT}} \boldsymbol{I}$ // *diagonal precision*
2: $\boldsymbol{U} \leftarrow \boldsymbol{0}$, $\boldsymbol{\Sigma} \leftarrow \boldsymbol{0}$ // *empty low-rank term*
3: **return** $(\boldsymbol{D}, \boldsymbol{U}, \boldsymbol{\Sigma})$

---

**add\_compute\_inv\_sum**

Inverts the precision according to the Woodbury identity.

**Require:** current $(\boldsymbol{D}, \boldsymbol{U}, \boldsymbol{\Sigma})$, diagonal $\boldsymbol{Q}$
1: $\boldsymbol{L} \leftarrow (\boldsymbol{Q} + \boldsymbol{D}_t^{-1})^{-1}$
2: $\boldsymbol{U}' \leftarrow \boldsymbol{D}_t^{-1} \odot \boldsymbol{U}_t$
3: $\boldsymbol{C}' \leftarrow \text{DIAG}((\text{DIAG}(\boldsymbol{C}_t) + \varepsilon \boldsymbol{I})^{-1}) + \boldsymbol{U}_t^{\top} \boldsymbol{U}'$
4: $\boldsymbol{D}_t \leftarrow \boldsymbol{L}$, $\boldsymbol{U}_t \leftarrow \boldsymbol{L} \odot \boldsymbol{U}'$, $\boldsymbol{C}_t \leftarrow (\boldsymbol{C}' - (\boldsymbol{U}')^{\top} \boldsymbol{U}_t)^{\dagger}$
5: **return** $(\boldsymbol{D}_t, \boldsymbol{U}_t, \boldsymbol{C}_t)$

---

**update\_mP**

Computes the vector matrix product of mean and the diagonal plus low-rank precision.

**Require:** current $(\boldsymbol{D}_t, \boldsymbol{U}_t, \boldsymbol{\Sigma}_t)$, vector $\boldsymbol{m}_t$
1: $\boldsymbol{m}_t \boldsymbol{P}_t \leftarrow \boldsymbol{D}_t \odot \boldsymbol{m}_t + \boldsymbol{U}_t \boldsymbol{C}_t \boldsymbol{U}_t^{\top} \boldsymbol{m}_t$
2: **return** $\boldsymbol{m}_t \boldsymbol{P}_t$

---

---

**add_low**

Updates the precision with the Jacobian and Hessian of the GGN. Computes the square root of the low-rank part and for each batch dimension of the Jacobian and Hessian. Then, uses the $r$-truncated SVD. Approximates the diagonal.

**Require:** current $(\boldsymbol{D}_t, \boldsymbol{U}_t, \boldsymbol{C}_t)$, Jacobians $\boldsymbol{J}_b$,
  Hessians $\boldsymbol{H}_b$, task scaling $\lambda_{\text{task}}$, rank $r$, $\varepsilon > 0$
1: $\boldsymbol{W}_0 \leftarrow \boldsymbol{U}_t \boldsymbol{C}_t^{1/2}$
2: **for** $b = 1, \dots, B$ **do**
3:    $\boldsymbol{W}_b \leftarrow \frac{1}{\sqrt{B}}(\sqrt{\lambda_{\text{task}}}\, \boldsymbol{J}_b)^\top (\boldsymbol{H}_b + \varepsilon \boldsymbol{I})^{1/2}$
4: $\boldsymbol{W} \leftarrow [\boldsymbol{W}_0 \; \boldsymbol{W}_1 \; \cdots \; \boldsymbol{W}_B]$
5: $(\tilde{\boldsymbol{U}}, \tilde{\boldsymbol{\Sigma}}) \leftarrow \text{TSVD}_r(\boldsymbol{W})$
6: $\boldsymbol{U}_t \leftarrow \tilde{\boldsymbol{U}}, \quad \boldsymbol{C}_t \leftarrow \text{DIAG}(\tilde{\boldsymbol{\Sigma}}^2)$
7: $\boldsymbol{D}_t \leftarrow \boldsymbol{D}_t + \text{APPROXIMATE\_DIAG}(\sqrt{\lambda_{\text{task}}}\, \boldsymbol{J}, \boldsymbol{H})$
8: **return** $(\boldsymbol{D}_t, \boldsymbol{U}_t, \boldsymbol{C}_t)$

---

**smooth**

Updates the mean based on the smoothing gain.

**Require:** mean $\boldsymbol{m}_t$, smoothing gain $\boldsymbol{G}_t$, next $\boldsymbol{m}_{t+1}$
1: $\boldsymbol{m}_t \leftarrow \boldsymbol{m}_t + \boldsymbol{G}_t(\boldsymbol{m}_{t+1} - \boldsymbol{m}_{t+1}^-)$
2: **return** $\boldsymbol{m}_t$

---

We implemented a class object DIAGLOWRANK with the functions mentioned above..

## C   Filtering and Smoothing with Diagonal plus Low-Rank Matrices

### C.1   Mathematical operations on diagonal plus low-rank matrices

In the following we describe how to perform mathematical operations on diagonal plus low-rank matrices $\boldsymbol{P} = \boldsymbol{D} + \boldsymbol{U}\boldsymbol{\Sigma}\boldsymbol{U}^\top$, where $\boldsymbol{D} \in \mathbb{R}^{D \times D}$ is a diagonal matrix, $\boldsymbol{U} \in \mathbb{R}^{D \times k}$ is a tall matrix, and $\boldsymbol{\Sigma} \in \mathbb{R}^{k \times k}$ is a dense matrix, with $k \ll D$. And importantly, all of these operations should be such that they maintain the low-rank structure of the matrix such that we never need to store the full matrix in memory, and they should be computationally efficient and scale at most linearly in $D$.

- **Addition with a diagonal matrix:** Adding a diagonal matrix to a diagonal plus low-rank matrix results in a diagonal plus low-rank matrix:

$$\left(\boldsymbol{D} + \boldsymbol{U}\boldsymbol{\Sigma}\boldsymbol{U}^\top\right) + \Lambda = \underbrace{\left(\boldsymbol{D} + \Lambda\right)}_{\boldsymbol{D}'} + \boldsymbol{U}\boldsymbol{\Sigma}\boldsymbol{U}^\top \tag{17}$$

- **Matrix inversion:** The inverse of a diagonal plus low-rank matrix can be computed efficiently using the Woodbury matrix identity:

$$\left(\boldsymbol{D} + \boldsymbol{U}\boldsymbol{\Sigma}\boldsymbol{U}^\top\right)^{-1} = \underbrace{\boldsymbol{D}^{-1}}_{\boldsymbol{D}'} + \underbrace{\boldsymbol{D}^{-1}\boldsymbol{U}}_{\boldsymbol{U}'} \underbrace{\left(-\left(\boldsymbol{\Sigma}^{-1} - \boldsymbol{U}^\top \boldsymbol{D}^{-1}\boldsymbol{U}\right)^{-1}\right)}_{\boldsymbol{\Sigma}'} \underbrace{\boldsymbol{U}^\top \boldsymbol{D}^{-1}}_{(\boldsymbol{U}')^\top} \tag{18}$$

- **Addition of two low-rank matrices:** Adding two diagonal plus low-rank matrices results in a diagonal plus low-rank matrix, but with increased rank $k' = k_1 + k_2$:

$$\left(\boldsymbol{D}_1 + \boldsymbol{U}_1\boldsymbol{\Sigma}_1\boldsymbol{U}_1^\top\right) + \left(\boldsymbol{D}_2 + \boldsymbol{U}_2\boldsymbol{\Sigma}_2\boldsymbol{U}_2^\top\right) = (\boldsymbol{D}_1 + \boldsymbol{D}_2) + \begin{bmatrix} \boldsymbol{U}_1 & \boldsymbol{U}_2 \end{bmatrix} \begin{bmatrix} \boldsymbol{\Sigma}_1 & 0 \\ 0 & \boldsymbol{\Sigma}_2 \end{bmatrix} \begin{bmatrix} \boldsymbol{U}_1^\top \\ \boldsymbol{U}_2^\top \end{bmatrix} \tag{19}$$

Alternatively, to keep the rank low we can perform a truncated singular value decomposition on the matrix square-root:

$$\tilde{\boldsymbol{U}}\tilde{\boldsymbol{\Sigma}}\tilde{\boldsymbol{V}}^\top = \text{tSVD}_k\left(\begin{bmatrix} \boldsymbol{U}_1\boldsymbol{\Sigma}_1^{1/2} & \boldsymbol{U}_2\boldsymbol{\Sigma}_2^{1/2} \end{bmatrix}\right) \tag{20}$$

and then approximate

$$\left(\boldsymbol{D}_1 + \boldsymbol{U}_1\boldsymbol{\Sigma}_1\boldsymbol{U}_1^\top\right) + \left(\boldsymbol{D}_2 + \boldsymbol{U}_2\boldsymbol{\Sigma}_2\boldsymbol{U}_2^\top\right) \approx \underbrace{(\boldsymbol{D}_1 + \boldsymbol{D}_2)}_{\boldsymbol{D}'} + \underbrace{\tilde{\boldsymbol{U}}}_{\boldsymbol{U}'}\underbrace{\tilde{\boldsymbol{\Sigma}}^2}_{\boldsymbol{\Sigma}'}\underbrace{\tilde{\boldsymbol{U}}^\top}_{(\boldsymbol{U}')^\top} \tag{21}$$

## C.2  Predict Step

Given a precision matrix $\boldsymbol{P}$ and a transition noise covariance $\boldsymbol{Q}$, the predict step in a Gaussian filter computes the predictive precision as

$$\boldsymbol{P}^- = \left(\boldsymbol{P}^{-1} + \boldsymbol{Q}\right)^{-1}. \tag{22}$$

Now if the precision is a diagonal plus low-rank matrix $\boldsymbol{P} = \boldsymbol{D} + \boldsymbol{U}\boldsymbol{\Sigma}\boldsymbol{U}^\top$, and the transition noise covariance $\boldsymbol{Q}$ is diagonal, we can show that the predictive precision is also diagonal plus low-rank. First, we apply the Woodbury matrix identity to the precision matrix:

$$\left(\boldsymbol{Q} + \boldsymbol{P}^{-1}\right)^{-1} = \left(\boldsymbol{Q} + \left(\boldsymbol{D} + \boldsymbol{U}\boldsymbol{\Sigma}\boldsymbol{U}^\top\right)^{-1}\right)^{-1} \tag{23}$$

$$= \left(\boldsymbol{Q} + \boldsymbol{D}^{-1} - \boldsymbol{D}^{-1}\boldsymbol{U}\left(\boldsymbol{\Sigma}^{-1} + \boldsymbol{U}^\top\boldsymbol{D}^{-1}\boldsymbol{U}\right)^{-1}\boldsymbol{U}^\top\boldsymbol{D}^{-1}\right)^{-1}. \tag{24}$$

Defining $\boldsymbol{D}' := \boldsymbol{Q} + \boldsymbol{D}^{-1}$, $\boldsymbol{U}' := \boldsymbol{D}^{-1}\boldsymbol{U}$, and $\boldsymbol{\Sigma}' := -\left(\boldsymbol{\Sigma}^{-1} + \boldsymbol{U}^\top\boldsymbol{D}^{-1}\boldsymbol{U}\right)^{-1}$, and applying the Woodbury matrix identity again, we get

$$\left(\boldsymbol{Q} + \boldsymbol{P}^{-1}\right)^{-1} = \boldsymbol{D}'^{-1} - \boldsymbol{D}'^{-1}\boldsymbol{U}'\left(\boldsymbol{\Sigma}'^{-1} + \boldsymbol{U}'^\top\boldsymbol{D}'^{-1}\boldsymbol{U}'\right)^{-1}\boldsymbol{U}'^\top\boldsymbol{D}'^{-1} \tag{25}$$

This shows that the predictive precision is also diagonal plus low-rank:

$$\boldsymbol{P}^- = \boldsymbol{D}^- + \boldsymbol{U}^-\boldsymbol{\Sigma}^-(\boldsymbol{U}^-)^\top, \tag{26}$$

with

$$\boldsymbol{D}^- := \boldsymbol{D}'^{-1} = \left(\boldsymbol{Q} + \boldsymbol{D}^{-1}\right)^{-1} \tag{27}$$

$$\boldsymbol{U}^- := \boldsymbol{D}'^{-1}\boldsymbol{U}' = \left(\boldsymbol{Q} + \boldsymbol{D}^{-1}\right)^{-1}\boldsymbol{D}^{-1}\boldsymbol{U}, \tag{28}$$

$$\boldsymbol{\Sigma}^- := -\left(\boldsymbol{\Sigma}'^{-1} + \boldsymbol{U}'^\top\boldsymbol{D}'^{-1}\boldsymbol{U}'\right)^{-1} = \left(\boldsymbol{\Sigma}^{-1} + \boldsymbol{U}^\top\boldsymbol{D}^{-1}\boldsymbol{U} - \boldsymbol{U}^\top\boldsymbol{D}^{-1}\left(\boldsymbol{Q} + \boldsymbol{D}^{-1}\right)^{-1}\boldsymbol{D}^{-1}\boldsymbol{U}\right)^{-1} \tag{29}$$

## C.3  Update Step (Diagonal Approximation)

We approximate the diagonal before the tSVD which otherwise discards the crucial diagonal information. We estimate the diagonal of the GGN with Hutchinson randomized estimator using $K$ Rademacher vectors $\boldsymbol{v} \in \{-1, +1\}^D$. Since for a matrix $\boldsymbol{A}$, it holds $\mathbb{E}[\boldsymbol{v} \odot \boldsymbol{A}\boldsymbol{v}] = \operatorname{diag}(\boldsymbol{A})$:

$$\boldsymbol{D}_t := \frac{1}{K}\frac{1}{B}\sum_{k=1}^{K}\sum_{b=1}^{B}\left(\boldsymbol{v}^{(k)} \odot \left(\boldsymbol{J}_b^\top \boldsymbol{H}_b \boldsymbol{J}_b \, \boldsymbol{v}^{(k)}\right)\right), \qquad \boldsymbol{v}^{(k)} \in \{\pm 1\}^d. \tag{30}$$

## C.4  Smoother Step

The standard Kalman smoother, or Rauch–Tung–Striebel smoother (Rauch et al., 1965), computes Gaussian posterior distributions

$$p(\boldsymbol{\theta}_t \mid \mathbb{D}_{1:T}) = \mathcal{N}(\boldsymbol{\theta}_t; \boldsymbol{m}_t^s, \boldsymbol{C}_t^s) \tag{31}$$

by iterating the following backward recursion, starting with the filtering distribution $\boldsymbol{m}_T^s = \boldsymbol{m}_T$ and $\boldsymbol{C}_T^s = \boldsymbol{C}_T$:

$$\boldsymbol{G}_t = \boldsymbol{C}_t \left(\boldsymbol{C}_t^-\right)^{-1}, \tag{32}$$

$$\boldsymbol{m}_t^s = \boldsymbol{m}_t + \boldsymbol{G}_t \left(\boldsymbol{m}_{t+1}^s - \boldsymbol{m}_{t+1}^-\right), \tag{33}$$

$$\boldsymbol{C}_t^s = \boldsymbol{C}_t + \boldsymbol{G}_t \left(\boldsymbol{C}_{t+1}^s - \boldsymbol{C}_{t+1}^-\right)\boldsymbol{G}_t^\top. \tag{34}$$

Now let us formulate the smoother step in terms of diagonal plus low-rank precision matrices. Let the filtering precision at time $t$ be $\boldsymbol{P}_t = \boldsymbol{D}_t + \boldsymbol{U}_t\boldsymbol{\Sigma}_t\boldsymbol{U}_t^\top$, the smoothing precision at time $t+1$ be $\boldsymbol{P}_{t+1}^s = \boldsymbol{D}_{t+1}^s +$

$U_{t+1}^s \Sigma_{t+1}^s (U_{t+1}^s)^\top$, and let $Q$ be diagonal. Recall that the predicted precision satisfies $P_{t+1}^- = \left(P_t^{-1} + Q\right)^{-1}$. Then, the smoothing gain $G_t$ is given by

$$G_t = C_t \left(C_{t+1}^-\right)^{-1} = P_t^{-1} P_{t+1}^- = P_t^{-1} \left(P_t^{-1} + Q\right)^{-1} = (I + QP_t)^{-1}. \tag{35}$$

Plugging in the diagonal plus low-rank form of the precision matrices, we get

$$G_t = \big( \underbrace{I + QD_t}_{D'} + \underbrace{QU_t}_{U'} \underbrace{\Sigma_t}_{\Sigma'} \underbrace{U_t^\top}_{V'^\top} \big)^{-1} \tag{36}$$

Applying the Woodbury matrix identity, we get

$$G_t = D'^{-1} - D'^{-1} U' \left(\Sigma'^{-1} + V'^\top D'^{-1} U'\right)^{-1} V'^\top D'^{-1}. \tag{37}$$

Therefore, the smoothing gain is diagonal plus low-rank $G_t = D_t^G + U_t^G \Sigma_t^G (V_t^G)^\top$, with

$$D_t^G := D'^{-1} = (I + QD_t)^{-1}, \tag{38}$$

$$U_t^G := D'^{-1} U' = (I + QD_t)^{-1} QU_t, \tag{39}$$

$$\Sigma_t^G := -\left(\Sigma'^{-1} + V'^\top D'^{-1} U'\right)^{-1} = -\left(\Sigma_t^{-1} + U_t^\top (I + QD_t)^{-1} QU_t\right)^{-1}, \tag{40}$$

$$\left(V_t^G\right)^\top := V'^\top D'^{-1} = U_t^\top (I + QD_t)^{-1}. \tag{41}$$

This concludes the first part: Computing the smoothed mean as in Equation (33) can be done efficiently as $G_t$ is diagonal plus low-rank.

The smoothing covariance/precision can again be approximated efficiently in a diagonal plus low-rank manner. We first re-write the smoothing covariance in terms of precisions, and re-order some terms to obtain an addition of two low-rank matrices:

$$(P_t^s)^{-1} = (P_t)^{-1} + G_t \left(\left(P_{t+1}^s\right)^{-1} - \left(P_{t+1}^-\right)^{-1}\right) G_t^\top \tag{42}$$

$$= (P_t)^{-1} + G_t \left(\left(P_{t+1}^s\right)^{-1} - (P_t)^{-1} - Q\right) G_t^\top \tag{43}$$

$$= (P_t)^{-1} - G_t (P_t)^{-1} G_t^\top + G_t \left(\left(P_{t+1}^s\right)^{-1} - Q\right) G_t^\top \tag{44}$$

$$= (I - G_t)(P_t)^{-1}(I - G_t)^\top + G_t \left(\left(P_{t+1}^s\right)^{-1} - Q\right) G_t^\top. \tag{45}$$

Then, since $P_t$ and $P_{t+1}^s$ are diagonal plus low-rank, their inverse is also diagonal plus low-rank (see Appendix C.1). Let $(P_t)^{-1} = D_t + U_t \Sigma_t U_t^\top$ and $(P_{t+1}^s)^{-1} = D_{t+1}^s + U_{t+1}^s \Sigma_{t+1}^s (U_{t+1}^s)^\top$. Then, the smoothing precision can be written as

$$(P_t^s)^{-1} = (I - G_t)\left(D_t + U_t \Sigma_t U_t^\top\right)(I - G_t)^\top + G_t \left(D_{t+1}^s + U_{t+1}^s \Sigma_{t+1}^s (U_{t+1}^s)^\top - Q\right) G_t^\top. \tag{46}$$

Similarly to before in Appendix C.1, we can write the smoothing precision as a matrix product $(P_t^s)^{-1} = WW^\top$, by defining $W$ as

$$W := \left[(I - G_t)D_t^{1/2} \quad (I - G_t)U_t \Sigma_t^{1/2} \quad G_t(D_{t+1}^s - Q)^{1/2} \quad G_t U_{t+1}^s (\Sigma_{t+1}^s)^{1/2}\right]. \tag{47}$$

Then, we can perform a truncated SVD on $W \approx U\Sigma V^\top$ to obtain a low-rank approximation of the smoothing covariance

$$(P_t^s)^{-1} \approx U\Sigma^2 U^\top. \tag{48}$$

A low-rank approximation of the precision follows again with the Woodbury matrix identity (see Appendix C.1).

| Dataset | x (input) | y (label) | d (domain) | Other metadata | Eval metric |
|---|---|---|---|---|---|
| **Camelyon17-wilds** | Patch from lymph node WSI | Binary: if central 32×32 has tumor | Hospital ID (5) | WSI ID | Avg. accuracy |
| **iWildCam2020-wilds** | Camera-trap photo | 182-way species | Camera trap ID | Sequence ID (burst) Date-time | Macro F1 |

Table 2: **Selected WILDS datasets**. Summary presenting inputs/labels/domains, metadata, and official evaluation metric.

# D    Experimental Details

## D.1    Datasets

The datasets that we used are MNIST (Deng, 2012), CAMELYON and IWILDCAM (Koh et al., 2021). Our implementation is in JAX (Bradbury et al., 2018).

In Section 5 we use *Permuted* and *Disjoint* MNIST. Next, we create two new settings within the CAMELYON and IWILDCAM datasets (please see Table 2) from the WILDS benchmark, a collection of datasets designed to address distribution shifts commonly encountered in real-world scenarios. These include domain generalization, where the objective is to generalize to unseen domains during training, and subpopulation shift, where class proportions differ.

CAMELYON19 is a dataset of histopathology images of lymph node sections, collected from five different hospitals in the Netherlands. The variations in population, in slide staining and image acquisition across hospitals make it a good candidate for testing the robustness of models to distribution shifts. The input $x$ is the 96x96 image and $y$ is a binary label indicating the presence of metastasis in the central 32x32, domain $d$ represents the hospital ID. The dataset holds 450 000 patches from 50 whole-slide images (WSI) with 10 WSI per hospital. The additional metadata is the WSI ID.

IWILDCAM is a dataset of camera traps monitoring wildlife. Different traps have various illumination, color, angle, and background, which makes it a good candidate for testing the robustness of models to distribution shifts. The input $x$ is the photo, $y$ is one of 182 animal species and $d$ is the camera trap ID. The dataset holds 203 029 images from 323 different camera traps across different countries in different parts of the world. The images are taken in bursts following trap activation so the images are also grouped into sequences. Therefore, the metadata is as follows camera trap ID, sequence ID and date-time. 35% of images are empty. There exists some label noise since some empty images might get an animal label if they belong to a sequence with that animal and some are labeled as empty. This poses a very interesting dataset for our continual learning setup. We create tasks by grouping the data by camera trap ID. We pre-filter that the camera ID has to have at least 1000 images.

### D.1.1    Continual learning task construction for Camelyon.

We adapted CAMELYON for a continual learning setup, expanding it with additional examples that are simple yet illustrative of dataset shifts, following as a guideline the work by Quionero-Candela et al. (2009). We believe they capture realistic scenarios e.g. brightness shifts due to different generations of machines, varying staining methods or imbalanced amount of data.

GRADUAL CAMELYON:  We apply a task-specific brightness change $\Delta_t$ to all pixels of all images i.e. $x_t = x + \Delta_t$. Next, we apply normalization with the mean and standard deviation shared for all tasks i.e. $(x_t - \mu_X)/\sigma_X$. Since each task uses a different brightness shift, it is not negated by the shared normalization operation (in a real-world application this could happen if the machine's manufacturer has some default normalization, but each hospital might have an individual shift, e.g. due to individual machine's degradation, varying light setup, etc.). In Figure 6 we show some exemplary images from each task.

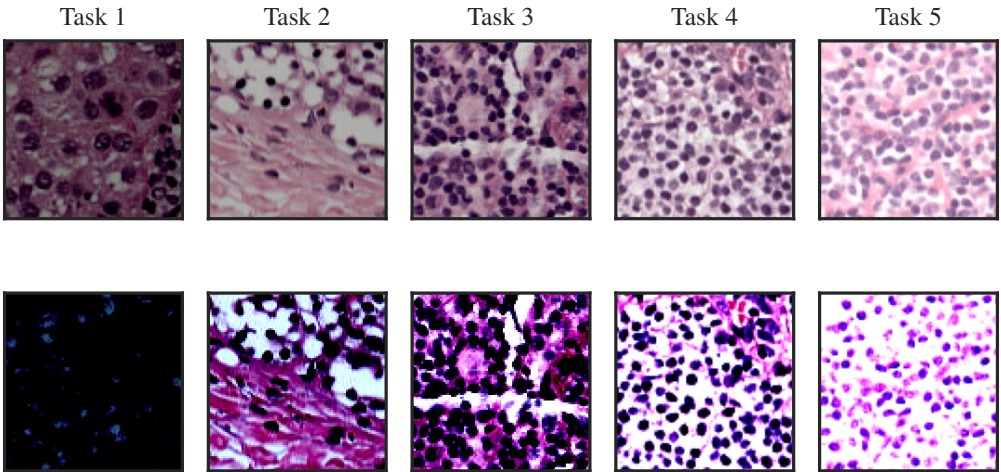

Figure 6: **Examples of the input data from** GRADUAL CAMELYON. (*Top*) To adjust the brightness, we apply a shift $x_t = x + \Delta_t$ (*bottom*) next, we normalize $(x_t - \mu_X)/\sigma_X$.

| Setting | Task | Images (train/test/valid) | Class 0 | Class 1 |
|---|---|---|---|---|
| GRADUAL CAMELYON | 0 | 5000 / 2000 / 2000 | 2528 / 991 / 1000 | 2472 / 1009 / 1000 |
| GRADUAL CAMELYON | 1 | 5000 / 2000 / 2000 | 2537 / 998 / 1010 | 2463 / 1002 / 990 |
| GRADUAL CAMELYON | 2 | 5000 / 2000 / 2000 | 2478 / 994 / 961 | 2522 / 1006 / 1039 |
| GRADUAL CAMELYON | 3 | 5000 / 2000 / 2000 | 2471 / 945 / 1038 | 2529 / 1055 / 962 |
| GRADUAL CAMELYON | 4 | 5000 / 2000 / 2000 | 2511 / 980 / 1024 | 2489 / 1020 / 976 |
| **Setting** | **Task** | **Images (train/test/valid)** | **Number of locations** | **Number of species** |
| LOCATIONSPLIT iWILDCAM | 0 | 28125 / 3523 / 3574 | 65 / 56 / 56 | 114 / 83 / 90 |
| LOCATIONSPLIT iWILDCAM | 1 | 31692 / 4057 / 3985 | 65 / 58 / 59 | 108 / 82 / 82 |
| LOCATIONSPLIT iWILDCAM | 2 | 36669 / 4645 / 4603 | 65 / 61 / 63 | 112 / 80 / 86 |
| LOCATIONSPLIT iWILDCAM | 3 | 38388 / 5091 / 5128 | 64 / 60 / 58 | 121 / 89 / 97 |
| LOCATIONSPLIT iWILDCAM | 4 | 26589 / 3553 / 3407 | 64 / 57 / 52 | 117 / 89 / 88 |

Table 3: **Data splits across tasks**. Per-task train/test/valid counts for $T = 5$.

### D.1.2  Continual learning task construction for iWildCam.

LOCATIONSPLIT iWILDCAM: We construct tasks by disjoint camera locations to emphasize low-level distribution shift. All unique camera IDs are ranked by the number of distinct species they observe (species richness), and then distributed *round-robin* into $T$ disjoint blocks: the first location goes to task 0, the second to task 1, ..., the $T$-th to task $T-1$, then the cycle repeats. Task $t$ contains all images from its assigned locations; no location appears in multiple tasks. Within each task, sequences are kept intact (no sequence split). We split sequences into train/val/test with an 80/10/10 ratio by sequence count (single-sequence locations are assigned to train). This yields zero location overlap by design, while species overlap depends on how widely species appear across locations (for $T = 5$ average overlap is 46.88% when quantified as the proportional overlap within the set of shared species and 61.844% when measured only in terms of species overlap). In Figure 7 we show some exemplary images from each task.

### D.2  Experimental Setup

Table 4 reports the values of all hyperparameters for each result presented in the paper. All of the experiments were run on local desktop with one NVIDIA GeForce RTX 2080 Ti with 11 GB memory, local laptop, Apple

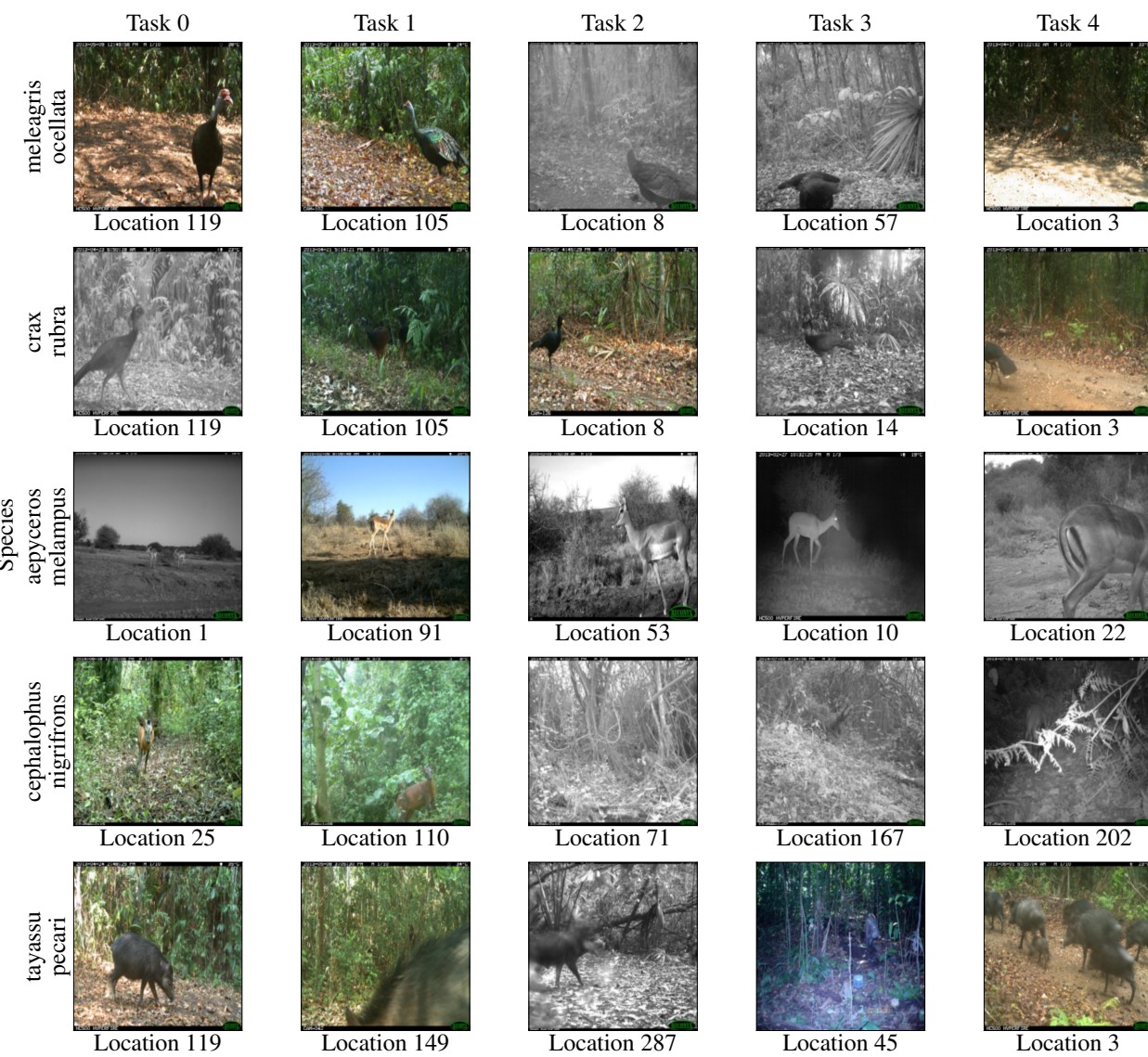

Figure 7: **Examples of the input data from** LOCATIONSPLIT IWILDCAM. Each column shows a different task and each row shows common species across tasks. We see that each species is shown in different, non-overlapping locations across tasks.

Table 4: **Experimental setup: MNIST**. The hyperparameters were tuned via an independent search on a grid on validation set.

| Experiment | *Permuted* MNIST | *Disjoint* MNIST |
|---|---|---|
| layers | | 2 |
| units | | 400 |
| epochs | | 10 |
| points | 60 000 | 12 000 |
| $\lambda$ | ours: $10^5$, diag EWC: $10^9$ OSLA: 10 | ours: $5 \cdot 10^4$, diag EWC: $9 \cdot 10^5$ OSLA: 10 |
| weight decay | 0 | $10^{-5}$ |
| seeds | | 8 |
| batch size | | 128 |
| Hessian batch size | | 128 |
| learning rate | 0.001 | 0.00001 |
| $\lambda_{\text{INIT}}$ | 0.0001 | 0.00001 |

Table 5: **Experimental setup: CAMELYON**. The hyperparameters were tuned via an independent search on a grid on validation set.

| Experiment | Gradual CAMELYON study of Q (Sections 5.1 and 5.3) | Gradual CAMELYON smoother (Section 5.2) | LocationSplit iWildCam comparison (Section 5.3) |
|---|---|---|---|
| layers | | 3 Conv 2 Dense | |
| units | | 32, (3x3) 16, (3x3) 4, (3x3) 8, 2 | |
| epochs | | 5 | |
| points | 5000 | 750 | all |
| $\lambda$ | ours: $10^5$ EWC: $10^6$ OSLA: 1 | $10^4$ | ours: 10 EWC: 0.01 OSLA: 1000 |
| weight decay | | 0 | |
| seeds | | 8 | |
| $Q$ | structured: $10^{-5}$ (C0 or C0 & C1) scalar: $10^{-6}$ | $10^{-5}$ | 0 |
| batch size | | 32 | |
| Hessian batch size | | 32 | |
| learning rate | | 0.0005 | |
| initial $\lambda$ | | 0.01 | |

M3 Max, with 64 GB memory, one NVIDIA 2080ti with 11 GB or one NVIDIA A100 with 40 GB. The runtime of each experiment ranged from 2 minutes to 150 minutes.

Table 6: **Computational requirements of each operation**. We provide computational, memory complexity of each step, where $D$ is the number of parameters, $r$ is the rank of the precision, $C$ is the number of classes, $B$ is the batch size, $i$ is the number of iterations in tSVD, $K$ is the number of Rademacher vectors. We report the steady-state time for each step in milliseconds, stored state and peak GPU memory for the experiment in Section 5.2. Measurements are from one seed.

| | Computational complexity | Memory complexity | Time [ms] | Stored state [MB] | Peak GPU [GB] |
|---|---|---|---|---|---|
| PREDICT | | | | | |
| ∘ inverse of sum (Algorithm 1 L1) | $\mathcal{O}(k^2 D + k^3)$ | $\mathcal{O}(kD + k^2)$ | 6.37 | 0.10 | 1.47 |
| ∘ update $\boldsymbol{m}_t^-$ (Algorithm 1 L1) | $\mathcal{O}(k^2 + kD)$ | $\mathcal{O}(k^2 + kD)$ | 2.02 | 0.10 | 1.47 |
| UPDATE | | | | | |
| ∘ compute GGN (Algorithm 1 L3) | $\mathcal{O}(C^2 + CD)$ | $\mathcal{O}(C^2 + CD)$ | 652.33 | 0.05 | 1.47 |
| ∘ approximate the diagonal L4 | $\mathcal{O}(KB(C^2 + CD))$ | $O(C^2 + CD)$ | 14.58 | 0.15 | 1.47 |
| ∘ square-root structure (Algorithm 1 L5) | $\mathcal{O}((r^2 + C^2 B)D)$ | $\mathcal{O}(D(B+1)k + k^2)$ | 52.68 | 0.10 | 1.47 |
| ∘ tSVD (Algorithm 1 L5) (Golub & Van Loan, 2013) | $\mathcal{O}(i(B+1)D)$ | $\mathcal{O}(r(B+1)D)$ | 7.99 | 0.10 | 1.47 |
| SMOOTH | | | | | |
| ∘ compute $\boldsymbol{G}_t$ (Eq. (12)) | $\mathcal{O}(k^2 D + k^3)$ | $\mathcal{O}(kD + k^2)$ | 4.32 | 0.10 | 1.97 |
| ∘ update $\boldsymbol{m}_t^s$ (Eq. (12)) | $\mathcal{O}(k^2 + kD)$ | $\mathcal{O}(k^2 + kD)$ | 1.60 | 0.10 | 1.97 |

Table 7: **Computational requirements of EWC and OSLA**. Let $d$ be the typical number of units in each layer and $B$ the mini-batch size, $T$ is the number of tasks, $N$ is the input dimension and $l$ is the number of intermediate layers. Measurements are from one seed.

| Operation | Computational complexity | Memory complexity | Time [ms] | Stored state [MB] | Peak GPU [GB] |
|---|---|---|---|---|---|
| **OSLA** | | | | | |
| ∘ compute $A$ | $\mathcal{O}(Bd(N + dl + C))$ | $\mathcal{O}(Bd(N + dl + C))$ | 175.17 | 0.32 | 1.46 |
| ∘ compute $G$ | $\mathcal{O}(Bd(N + dl + C))$ | $\mathcal{O}(Bd(N + dl + C))$ | 79.18 | < 0.01 | 1.46 |
| ∘ create blocks | $\mathcal{O}(B(N^2 + d^2 l + C^2))$ | $\mathcal{O}(TB(N^2 + d^2 l + C^2))$ | 0.57 | 0.32 | 1.46 |
| **EWC** | | | | | |
| ∘ compute the diagonal | $\mathcal{O}(BD)$ | $\mathcal{O}(BD)$ | 2.33 | 0.02 | 1.46 |
| ∘ update mean | $\mathcal{O}(D)$ | $\mathcal{O}(D)$ | 2.07 | 0.15 | 1.46 |

### D.3 Practical Guidance for Smoothing and Hyperparameters

Backward smoothing is expected to help when earlier task-specific models can benefit from later related tasks, but a single shared model is not optimal for all tasks. This can occur in low-data regimes or when tasks are related but still require task-specific parameter estimates. However, smoothing depends on the transition model and the chosen process noise $\boldsymbol{Q}$; if these are chosen incorrectly, smoothing may harm performance.

For the curvature batch size, in our experiments very small values proved to be detrimental, while $B \geqslant 4$ already gives comparable trends. We use $B = 32$ by default. For the rank $k$, we use $k = C$ by default. The regularization strength $\lambda$, as in other regularizer-based methods, needs to be tuned on validation data.

The process noise $\boldsymbol{Q}$ encodes prior knowledge about parameter change between tasks. It can be set layer-wise, e.g. larger in early layers for low-level shifts such as brightness, contrast, or staining, and larger in later layers for semantic or label-space shifts. Its magnitude controls the expected amount of parameter change; in our experiments, we scale it by $Q_{\text{val}} \cdot \text{mean}(\theta_l)^2$ for layer $l$. More automated choices could tie $\boldsymbol{Q}$ to task-discrepancy measures, such as intermediate activations or gradients, which we leave for future work. In practice, we treat $\lambda$, $k$, and the scale of $\boldsymbol{Q}$ as validation-selected hyperparameters.

### D.4 Experimental Results

We provide additional experimental results below.

Table 8: **Measured method-level profiler requirements**. We report steady-state time in milliseconds, measured stored array state and absolute peak process GPU memory for one seed.

| Method | Train step [ms] | Train state [MB] | Train peak GPU [GB] | Post-update [ms] | Post state [MB] | Post peak GPU [GB] | Run peak GPU [GB] |
|---|---|---|---|---|---|---|---|
| LR-LGF | 6.56 | 0.17 | 1.47 | 3211.55 | 0.61 | 1.47 | 1.97 |
| OSLA | 20.75 | 2.14 | 1.46 | 19595.22 | 2.50 | 1.46 | 1.46 |
| EWC | 11.64 | 0.32 | 1.46 | 3436.11 | 0.32 | 1.46 | 1.46 |

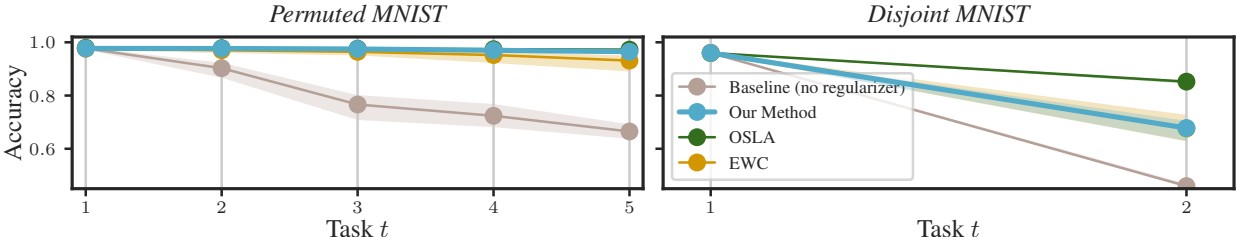

Figure 8: **Comparing our diagonal plus low-rank GGN approximation to other regularizers**. Mean performance of the current model on all previously encountered tasks (shaded areas are the min/max across 8 seeds). For *Permuted* and *Disjoint MNIST* we observe that our diagonal plus low-rank GGN approximation (—) leads to significantly lower rates of forgetting compared to no regularization (—). It also tends to be slightly better than EWC (—) and comparable to OSLA (—).

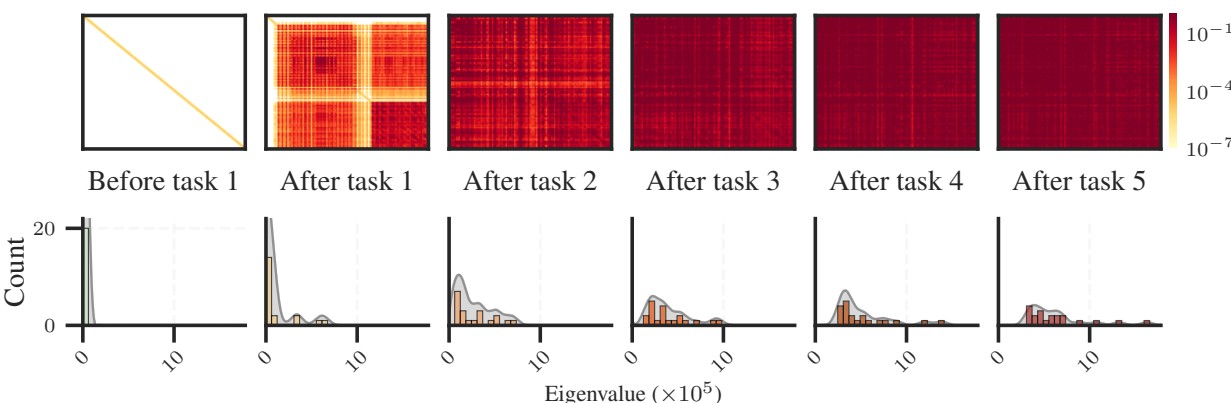

Figure 9: **The low-rank approximation across training (using rank k = 20).** *(Top)* Our rank $k = 20$ approximation to the precision matrix across tasks $t$ on *Permuted* MNIST (see Fig. 5 for $k = 10$). For better readability, values below $10^{-7}$ are shown in white. *(Bottom)* Histograms of the eigenvalues of the approximation's low-rank part. With growing $t$, the eigenvalues increase in magnitude, indicating larger certainty and less flexibility.

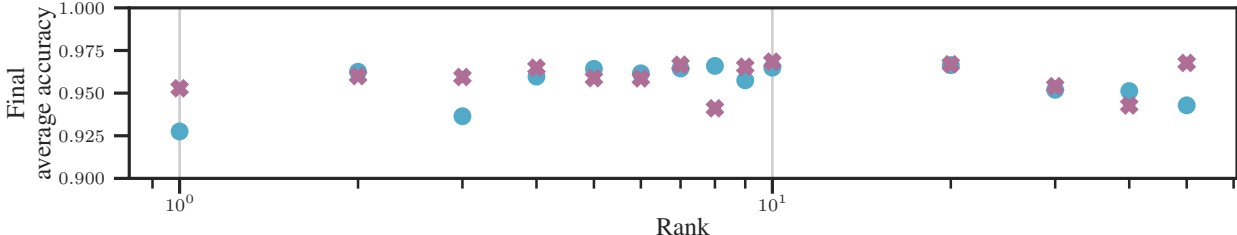

Figure 10: **Comparison of final average accuracy on *Permuted* MNIST for different ranks $k$ of our diagonal plus low-rank approximation**. We show the final average accuracy for two different seeds (● and ✖) when using varying ranks for the low-rank part. We observe that already a very low rank $k \geqslant 2$ provides a competitive performance. This indicates that a low rank is sufficient to provide a meaningful yet cheap curvature approximation. Increasing the rank beyond $\approx 10$ (the number of classes) offers little additional benefit and can even slightly degrade performance.

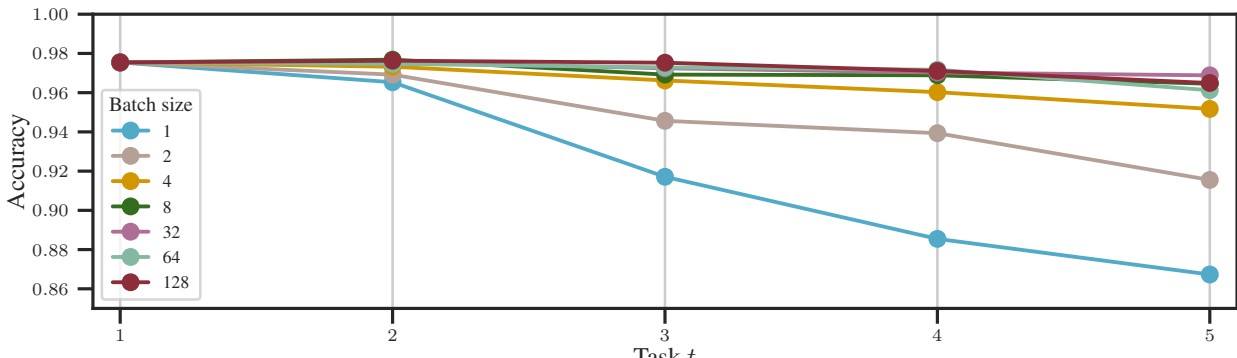

Figure 11: **The effect of the Hessian's batch size on the performance**. Using larger batch sizes to compute the GGN approximation of the Hessian help with the curvature estimation. Consequently, larger batch sizes also tend to improve the performance across tasks. However, we observe that a relatively modest and thus cheap batch size of 8 is sufficient to provide competitive performance.

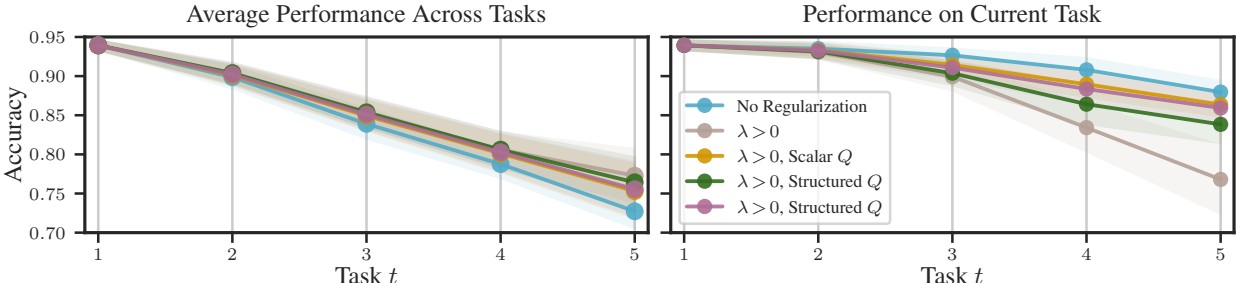

Figure 12: **The effect of $Q$ on the average and current task's performance (across seeds).** *(Left)* Looking at the average performance across currently observed tasks, we see that without regularization (──), performance significantly drops. With regularization (and possibly $Q > 0$) (e.g. ──, ──, ──), we can boost the average performance to roughly similar levels. *(Right)* Crucially, adding $Q > 0$ allows us to boost the performance on the current task, while maintaining a similar average performance. Adding a structured $Q$ (here, one that specifically targets the first two convolutional layers) (see Section 5.1), tends too perform slightly stronger in terms of average performance across our experiments. Shaded areas show $\pm$ one standard deviation across 8 random seeds.

