# OpenReview forum: "Low-Rank Filtering & Smoothing for Sequential Deep Learning"
_TMLR — Accepted by TMLR_

### Review · Reviewer_sbH4 · 2026-04-29

**Summary Of Contributions:**

This work formulate sequential learning (continual learning) from a Bayesian Learning perspective by treating neural network parameters as states. This formulation allows for a Bayesian Filtering view on neural network training on sequential tasks, enabling noise tracking over the neural network parameters, which was previously intractable for regularized sequential training approaches. This new formulation has also enabled some other useful techniques such as filtering based regularizer, task-specific noise scheduling and backward smoothing. The authors also provide a low rank approximation method to overcome the computation burden for the filtering problem. Simulation supports the effectiveness of this new training view.

Strength:
1. The filtering view for continual learning is interesting and makes sense.
2. The task specific noise Q sounds interesting and I can see some potential here to combine it with domain adaptation/distribution drift methods.
3. The filtering formulation provides genuine new angles and benefits to continual learning problem as demonstrated by the simulations.

Weakness:
1. Although most of the filtering formulation makes intuitive sense, there is a lack of theoretical justification on several details such as the quality of GGN approximation to the Hessian, and why and how good will backward smoothing work. Consequently, a comprehensive simulation should be provided in the absent of theoretical analysis. However, ablation studies such as the role of Q and backward smoothing are only done in one specific setting on one dataset.
2. Some of the method choices lack proper explanation or justification. For example, GNN approximation of Hessian is described, but term 2 in Equation 1 is simply omitted without clear explanation.
3. The filtering matrices in Section 3.2 are rather abruptly introduced, even that these are common notations in filtering literature, having them formally written and defined would smooth out the flow of the paper.
4. The task specific nature of Q makes sense, but the topic is not really explored in depth. Currently, the only settings to reflect task adaptation is differentiating between Q=0 vs Q>0, but in practice, task relationships are usually much more nuanced than this.

**Audience:**

Yes

**Audience Explanation:**

Both filtering and continual learning should be of interests to TMLR audience.

**Broader Impact Concerns:**

I do not see any concerns.

**Claims And Evidence:**

Yes

**Claims Explanation:**

I think the current simulation results all support the claim made by the authors, the only concern here is that the simulations are rather small in scale (number of datasets), so the evidence is not strong.

**Requested Changes:**

1. Theoretical analysis or explanation with reference or extra simulation to further support the methods of choice, such as the GGN approximation (Hessian approximation error) and backward smoothing should be added, to help the readers understand why they are good choice for neural network learning.
2. A discussion on how to actually make Q task specific should be provided. Even though that a complete noise scheduler Q based on things such as distribution discrepancy across task datasets is not needed, the authors should at least strengthen the explanation beyond 0 vs non-0 to actually support the task specific claim.
3. The matrix form Gaussian filtering formulation needs to be added ahead of Section 3.2 to make it easier for audience that are not on top of filtering literature to follow.

---

> ### Author Response · Authors · 2026-05-21
>
> We thank the reviewer for the time and helpful feedback. We appreciate that the reviewer views the work as interesting, with genuine new angles and benefits for continual learning, supported by simulations.
>
> > 1. [...] explanation with reference [...] the GGN approximation (Hessian approximation error) and backward smoothing should be added.
>
> We agree that the current draft does not sufficiently explain why the GGN approximation is appropriate in our setting.
>
> The Hessian decomposes due to the split between the model and the loss function:
> $\mathbf{H}(\boldsymbol{\theta}) = \frac{\partial^2 \mathcal{L}}{\partial \boldsymbol{\theta}^2} = \frac{\partial f}{\partial \boldsymbol{\theta}} \frac{\partial^2 \mathcal{L}}{\partial f^2} \left(\frac{\partial f}{\partial \boldsymbol{\theta}}\right)^{\top} + \frac{\partial^2 f}{\partial \boldsymbol{\theta}^2} \frac{\partial \mathcal{L}}{\partial f} := \mathbf{J} \hat{\mathbf{H}} \mathbf{J}^{\top} + \frac{\partial^2 f}{\partial \boldsymbol{\theta}^2} \frac{\partial \mathcal{L}}{\partial f}.$
>
> This gives two terms: the generalized Gauss-Newton term and a residual term involving second derivatives of the network outputs with respect to the parameters. The first term carries curvature information from the loss, while the second contains curvature of the model. If the loss is convex in $f$, the GGN term is positive semi-definite. The residual term may change the definiteness of the full Hessian. Therefore, by using the GGN, we obtain a stable positive semi-definite curvature approximation that can be used for posterior covariance estimation.
>
> We updated the introduction of this approximation and pointed to Dangel et al. (2022) (please see the Background section), who compare the full-batch GGN and full-batch Hessian (Figure 3), and study the impact of mini-batches (Figure 4). They suggest that this simplification leaves the curvature’s top eigenspace largely unaffected.
>
> As for backward smoothing, we follow the Rauch-Tung-Striebel smoother; it is a consequence of the Gaussian state-space formulation. The current manuscript states the recursive smoothing equations. The application of this smoother to neural-network sequential learning is one of the contributions of this work. We showcase the benefits in Figure 4, which highlights potential use cases in low-data settings where we cannot access the data again but can still improve the initial task-specific models. We added a short explanation of when smoothing is expected to help (please see Appendix D3): namely, when earlier task-specific models can benefit from later related tasks, but a single shared model is not optimal. If the transition model or $\mathbf{Q}$ are badly chosen, smoothing may harm performance.
>
> > 2. A discussion on how to actually make Q task specific should be provided.
>
> The noise matrix $\mathbf{Q}$ describes prior knowledge about parameter change from one task to the next one. Several practical strategies for setting it include:
>
> - Layer-based $\mathbf{Q}$: in our setting, we know that mostly low-level shifts occur, such as brightness, contrast, or staining changes. For such tasks, we would set $\mathbf{Q}$ for lower layers to be $>0$. For semantic or label-space shifts, larger noise may instead be assigned to later layers.
>
> - Magnitude of $\mathbf{Q}$: larger values indicate larger expected changes between tasks. In practice, we scale the chosen value of $\mathbf{Q}$ by the squared mean of the layer parameter values, i.e. $Q_{\mathrm{val}} \cdot \mathrm{mean}(\mathbf{\theta}_i)^2$.
>
> - Automated $\mathbf{Q}$ selection: if some current-task data or summary statistics are available, one could measure task discrepancy and connect it to the value of $\mathbf{Q}$, e.g. using gradients, intermediate activations, or other measures. We see this as an interesting direction for future work.
>
> Please see Appendix D3 for the changed manuscript.
>
> > 3. The matrix form Gaussian filtering formulation needs to be added ahead of Section 3.2 [...]
>
> We have added the text below before section 3.2.
>
> In dense matrix form, the computational bottlenecks of the above recursion are now explicit. The predict step requires forming the predicted covariance $\mathbf{C}_t^{-} =  \mathbf{C}^{t-1}+ \mathbf{Q} \in \mathbb{R}^{D \times D}$, whereas the update step requires computing the task Hessian and updating it with the previous task's covariance, $\mathbf{C}_t = (\lambda H_t + (\mathbf{C}_t^-)^{-1})^{-1}$, where $H_t = \nabla^2 \mathcal{L}(\boldsymbol{\theta}, \mathcal{D}_t)$ evaluated at $\boldsymbol{\theta}=\mathbf{m}_t$. Thus, although the Laplace-Gaussian filter has a simple matrix formulation, this remains impractical for deep learning due to the dense covariance and Hessian matrices being of prohibitive size. We resolve this with low-rank approximations.

---

### Review · Reviewer_9rxq · 2026-05-02

**Summary Of Contributions:**

The paper reframes sequential deep learning as approximate Bayesian inference in weight space. Concretely, it models neural-network parameters as latent states in a Gaussian state-space model, uses Laplace–Gaussian filtering to update the posterior after each task, introduces process noise (Q) as a way to encode prior beliefs about how tasks are related, and applies backward smoothing so that earlier task-specific models can benefit from later tasks without revisiting data. To make this practical, it derives a diagonal-plus-low-rank precision approximation based on the generalized Gauss–Newton matrix and truncated SVD, leading to the proposed LR-LGF method. Empirically, the paper studies three things: whether structured (Q) improves plasticity on a constructed domain-shift benchmark, whether smoothing helps earlier task-specific models, and whether the low-rank regularizer is competitive with EWC and OSLA on four continual-learning benchmarks.

Key strengths are the clean Bayesian framing, the genuinely interesting use of smoothing for rehearsal-free backward information transfer, and solid empirical support for the low-rank regularizer through multi-benchmark comparisons with 8 seeds and validation-based hyperparameter tuning. The main weaknesses are that the most novel capabilities—structured (Q) and smoothing—are demonstrated mainly on one custom benchmark family, practical selection of (Q) is not yet very well operationalized, scalability is limited when the output dimension (C) is large, and the manuscript itself notes that part of the diagonal-plus-low-rank machinery follows the same fundamental steps as prior low-rank filtering work, so the clearest novelty is the continual-learning formulation plus smoothing rather than the matrix approximation alone.

**Audience:**

Yes

**Audience Explanation:**

This paper sits comfortably within TMLR’s stated scope: it proposes a new algorithmic and analytical framework, and it backs it with both derivations and empirical validation. Even if the contribution is somewhat specialized, it should interest readers working in continual learning, Bayesian deep learning, approximate second-order methods, rehearsal-free adaptation, and sequential / online learning. The smoothing result is especially notable because it offers a principled way to improve earlier task-specific models without replaying data, which is a meaningful angle for privacy-constrained or storage-constrained settings.

I do not think the paper is broadly impactful enough, in its current form, to read as a “featured” or unusually high-significance TMLR paper. But TMLR’s bar is not “broad significance to everyone”; it is whether at least some of the journal’s audience would want to know the findings. For the communities above, the answer is clearly yes.

**Broader Impact Concerns:**

I do not see a major broader-impact concern that would, by itself, require a substantial Broader Impact Statement for acceptance. This is primarily a methodological paper.

**Claims And Evidence:**

Yes

**Claims Explanation:**

For the paper’s main, properly scoped claims, the evidence is mostly convincing. The methodology is developed clearly: the manuscript specifies the state-space model, derives the predict and update steps for Laplace–Gaussian filtering, derives backward smoothing, and then shows how the relevant matrix operations can be implemented efficiently with diagonal-plus-low-rank structure. The empirical section also covers the right layers of evidence: internal demonstrations for structured (Q), a targeted experiment for smoothing, and broader benchmark comparisons for the low-rank curvature approximation. On the main benchmark comparison, LR-LGF with (Q=0) outperforms EWC on all four reported benchmarks and matches or exceeds OSLA, while the smoothing experiment reports consistent gains across seeds and tasks, including a task-2 improvement from 70.5% to 75.9%.

My main reservation is not that the claims are unsupported, but that a few of them should be scoped carefully. The evidence for the low-rank regularizer is broader than the evidence for the new “task-relationship prior via (Q)” and “backward smoothing for task-specific models” capabilities. Those two points are demonstrated mainly on Gradual CAMELYON and a reduced-data variant, with a manually structured prior motivated by brightness shifts. So I believe the core technical claims are supported, but some of the broader application language—especially around privacy-critical deployment or general applicability of structured (Q)—would be stronger if phrased as promising evidence rather than fully general validation.

**Requested Changes:**

The changes below would improve the paper; the first two are the closest to critical because they affect claim scope and positioning.

- Critical / near-critical: Tighten the scope of the strongest claims around structured (Q) and smoothing. Either add one more qualitatively different experiment for each capability, or explicitly state that these are proof-of-concept demonstrations rather than broad real-world validation.

- Critical / near-critical: Clarify novelty relative to prior low-rank filtering work. The paper already states that its diagonal-plus-low-rank implementation follows the same fundamental steps as earlier work; the manuscript should therefore emphasize more sharply that the main novelty is the Laplace–Gaussian continual-learning formulation, the role of (Q) for task relationships, and the smoother.
Strengthening: Add direct wall-clock and memory comparisons against EWC and OSLA in the main paper. The appendix gives complexity tables and timing for the proposed method, but a more direct practical comparison would make the efficiency claim easier to judge.

- Strengthening: Provide more actionable guidance for choosing (Q), rank (k), Hessian batch size, and (\lambda). The ablations are useful, and the paper notes that setting dynamics parameters can be difficult in practice, but readers would benefit from a short “default recipe” section.

- Strengthening: Expand the discussion of scalability. The paper appropriately notes that the method becomes less attractive when the output dimension (C) is large; I would like one more paragraph discussing what the authors expect to happen on larger models / larger label spaces and when they would recommend K-FAC, block-diagonal, or last-layer alternatives instead.

- Strengthening: If the implementation is intended to be part of the reproducibility package, include a code link or an explicit release plan in the submission materials, consistent with the paper’s open-source implementation claim.

---

> ### Author Response · Authors · 2026-05-21
>
> We thank the reviewer for the time and helpful feedback. We appreciate the evaluation as a clearly developed methodology with solid empirical support and notable smoothing results.
>
> > Critical / near-critical: Tighten the scope of the strongest claims around structured $\mathbf{Q}$ and smoothing.
>
> We agree with this point. The experiments showcasing the benefits of $\mathbf{Q}$ and the backward smoother should be viewed as proof-of-concept demonstrations rather than broad real-world validation. We revised the manuscript to state this more explicitly (please see Limitations section).
>
> > Critical / near-critical: Clarify novelty relative to prior low-rank filtering work.
>
> Our main contributions focus on the Laplace-Gaussian formulation for continual learning, showcasing the benefits of domain knowledge prior via $\mathbf{Q}$, as well as, backward smoothing, beyond the utilization of the diagonal plus low-rank approximation. The diagonal-plus-low-rank approximation is the computational mechanism that makes these filtering and smoothing operations feasible for neural networks. We revised the manuscript (please see Related Work section).
>
> > Strengthening: Add direct wall-clock and memory comparisons against EWC and OSLA in the main paper.
>
> We compared the costs of each step of LR-LGF in Table 6 (updated), similarly for the other methods in Table 7 (updated) and added a joint summary per method in Table 8 (new).
>
> > Strengthening: Provide more actionable guidance for choosing $\mathbf{Q}$, rank $k$, Hessian batch size, and $\lambda$.
>
> In our current setting, very small curvature batch sizes are detrimental, but batch sizes of 4 and larger already lead to comparable trends. We use $B=32$ by default. For the rank, $k$ controls how much curvature structure is retained: small ranks already capture useful parameter interactions in our experiments, while larger ranks should be used only when validation performance improves enough to justify the extra memory and computation. We use $k=C$ by default.
>
> The regularization strength $\lambda$, as in other regularizer-based methods, requires hyperparameter tuning.
>
> The noise matrix $\mathbf{Q}$ describes prior knowledge about parameter change from one task to the next one. Several practical strategies for setting it include:
>
> - Layer-based $\mathbf{Q}$: in our setting, we know that mostly low-level shifts occur, such as brightness, contrast, or staining changes. For such tasks, we would set $\mathbf{Q}$ for lower layers to be $>0$. For semantic or label-space shifts, larger noise may instead be assigned to later layers.
>
> - Magnitude of $\mathbf{Q}$: larger values indicate larger expected changes between tasks. In practice, we scale the chosen value of $\mathbf{Q}$ by the squared mean of the layer parameter values, i.e. $Q_{\mathrm{val}} \cdot \mathrm{mean}(\mathbf{\theta}_i)^2$.
>
> - Automated $\mathbf{Q}$ selection: if some current-task data or summary statistics are available, one could measure task discrepancy and connect it to the value of $\mathbf{Q}$, e.g. using gradients, intermediate activations, or other measures. We see this as an interesting direction for future work.
>
> In practice, we treat $\lambda$, $k$, and the scale of $\mathbf{Q}$ as validation-selected hyperparameters. Please see Appendix D3 for the updated text.
>
> > Strengthening: Expand the discussion of scalability.
>
> The method is most practical when $BC$ is moderate and the chosen rank satisfies $k \ll D$. This is the case for the image-classification benchmarks considered here, where $C$ is small. In contrast, for problems with very large output spaces, the $BC$ factor can become prohibitively large. In such settings, more suitable variants would be: last-layer-only approximations when most uncertainty is expected near the classifier head; block-diagonal or layer-wise approximations when full-network interactions are too expensive e.g. K-FAC-style curvature. Therefore, LR-LGF should be viewed as a full-network curvature method for moderate-output continual-learning problems, positioned between diagonal methods such as EWC and denser second-order Bayesian filters. Please see the Limitations section for the updated text.
>
> > Strengthening: If the implementation is intended to be part of the reproducibility package, include a code link or an explicit release plan in the submission materials, consistent with the paper’s open-source implementation claim
>
> We will clarify the release plan.
> > Code release: We will provide the code and all scripts necessary to reproduce the results and figures upon acceptance.

---

### Review · Reviewer_2RrQ · 2026-05-08

**Summary Of Contributions:**

In this work the authors propose a method for continual learning based on an iterative Bayesian approach. They use a Gaussian approximation to define the Bayesian likelihood around a minimum using the generalized Gauss Newton matrix. This, combined with a prior, gives a regularizer for learning. This regularizer is built off of the geometry of the previous task in the sequence, and the authors give a derivation of how to use Bayesian assumptions to iteratively update the regularizer. They also discuss ways to make this iteration tractable using a diagonal-plus-low-rank approximation to the generalized Gauss Newton (and the resulting regularizing matrix). The authors also describe a smoothing procedure to give a task-specific model which incorporates information from all the models. The paper concludes with some experiments which suggest that the method may provide improvements over other continual learning algorithms.

**Audience:**

Yes

**Audience Explanation:**

I am not an expert in continual learning, but the overall idea of the iterative Bayesian approach seems like it would be interesting generally. Overall the derivation of the method is clean and well-presented and would be a good addition to the literature.

**Claims And Evidence:**

Yes

**Claims Explanation:**

The overall derivation of the method, modulo the concern below, seems correct. The experiments seem reasonable to me, but I am not an expert in continual learning and don't know if the chosen datasets and baseline algorithms were the correct ones to compare to.

One key early claim in the paper is not correct; the authors use equation (1), which decomposes the GGN into a product involving a C x C matrix (where C is the output dimension) to argue that the GGN is low rank (rank at most C). This is true for a single datapoint, but is not true when averaged over the whole dataset. The overall storage using the single-datapoint decomposition goes as O(BDC), where B is the batch size.

Relatedly, this means a lot of complexity/computational cost is potentially hidden in the compression of W described between 10 and 11. Indeed, in the setting of this paper B does not represent a training minibatch size, but the **dataset size**, which brings the whole low-rank framework, as well as computational tractability, in question for any interesting workloads. Without more detailed discussion and analysis of this point in the context of training practical models (for compute, memory, and performance), the current form of the paper feels misleading about the potential and pitfalls of the method.

Update post discussion: The authors have improved the main text by more explicitly discussing these points. This has addressed my concerns, and I have switched "claims and evidence" to "yes".

**Requested Changes:**

A more detailed discussion of the issues around dataset size, the complexity of the method, and the usefulness of the method as dataset size and complexity increases.

Some of the assumptions in the derivation require the loss to be from the exponential family; this should be more explicitly stated at the start.

Grid lines on figures 2 and 3 would greatly help in understanding the quantitative benefits of the model.

There may be connections between the proposed method and the curvature regularization literature. Specifically, sharpness aware minimization (SAM, https://arxiv.org/abs/2010.01412), particularly the penalty-SAM described [here](https://proceedings.neurips.cc/paper_files/paper/2024/hash/ee3ce0121939f42098cdefd3ea025bf1-Abstract-Conference.html) (more specifically, the "logit-PSAM" described [here](https://arxiv.org/pdf/2502.02407)). The main difference is that these methods use Hessian/GN gradient products where the Hessian is from the current point, while the proposed method uses something like GN-parameter products, for GN from the previous model (when Q = 0). Adding connections to the literature may strengthen the paper.

---

> ### Author Response · Authors · 2026-05-21
>
> We thank the reviewer for the time and helpful feedback.
> > A more detailed discussion of the issues around dataset size, the complexity of the method, and the usefulness of the method as dataset size and complexity increases.
>
> We agree that the current draft should discuss this more explicitly. In our implementation, we estimate the curvature using one batch of size $B$, and then truncate the resulting representation to rank $k$. In all our experiments, we use $k=C$. We distinguish between the training minibatch used for optimization and the curvature batch used to estimate the GGN. The curvature batch size $B=32$ is a fixed hyperparameter and is not the full dataset size. Using the full dataset would give a more accurate Laplace curvature estimate but would not be practical. We study the impact of the curvature batch size $B$ in Figure 11. In the paper, we also state the computational and memory constraints in the updated Table 6. We compare the costs to other methods in the updated Table 7 and add a new method-level comparison in Table 8. Additionally, please see the Limitations section for the updated text.
>
> To clarify, we construct a matrix $W_t$ of size $D \times (k+BC)$, and then perform truncated SVD on it.
> > Some of the assumptions in the derivation require the loss to be from the exponential family; this should be more explicitly stated at the start.
>
> We revised the manuscript. The likelihood interpretation in Eq. 2b is exact when the loss corresponds to a negative log-likelihood, as in common exponential-family observation models. The GGN is positive semi-definite whenever the loss is convex in the network outputs, as is the case for the standard losses considered here. Please see the Backgorund section and Section 3 for the updated text.
>
> >Grid lines on figures 2 and 3 would greatly help in understanding the quantitative benefits of the model.
>
> We updated the figures to include grid lines.
>
> > There may be connections between the proposed method and the curvature regularization literature. Specifically, sharpness aware minimization (SAM, https://arxiv.org/abs/2010.01412), particularly the penalty-SAM described here (more specifically, the "logit-PSAM" described here). The main difference is that these methods use Hessian/GN gradient products where the Hessian is from the current point, while the proposed method uses something like GN-parameter products, for GN from the previous model (when $\mathbf{Q}=0$). Adding connections to the literature may strengthen the paper.
>
> We agree that this is a useful connection. SAM seeks parameters whose neighborhoods have low loss, commonly formulated through the minimax objective $\min_{\theta} \max_{\|\epsilon\| \leq \gamma} L(\theta+\epsilon)$, where $\epsilon$ is a perturbation of the parameters. Related SAM variants, including penalty-SAM and logit-PSAM, study how curvature-related quantities affect sharpness regularization.
>
> The main distinction is that SAM-style methods use curvature information at the current point to encourage flatness and improve generalization. In contrast, our regularizer uses curvature accumulated from previous tasks. When $\mathbf{Q}=0$, it penalizes movement in directions important for previous tasks, as measured by the previous task's GGN. Thus, SAM-style methods primarily target generalization through sharpness control, whereas our method targets the stability-plasticity trade-off in sequential learning. We revised the Related Work section to make this connection and distinction clear (please see Appendix).

---

> > ### Comment · Reviewer_2RrQ · 2026-05-21
> > **Response to updates**
> >
> > I thank the authors for their updates. Most of my concerns have been addressed; however, the section on Laplace approximations (page 3 in the current draft) still fails to mention how the batch size affects the rank of the GGN. In my opinion this is a crucial point and must be addressed for acceptance.

---

> ### Author Response · Authors · 2026-05-21
>
> We thank the reviewer for the prompt response and for pointing this out. We agree that the rank statement in the GGN discussion should be made more precise.
>
> For a single data point, the corresponding GGN contribution has rank at most $C$. For the full dataset of size $N$, assuming $NC < D$, the GGN's rank is $\operatorname{rank}(\mathrm{GGN}) \leq NC.$
>
> In our setting, however, we do not construct the full-dataset GGN. We estimate the curvature using one batch, i.e. for $N = MB$, we set $M=1$, so the relevant rank is $ \min(D,BC)$.
> Thus, $B$ denotes the curvature batch size, not the dataset size. In our experiments, we use $B=32$.
>
> This is consistent with the construction in Section 3.2. The uncompressed factor $W_t$ contains the previous compressed precision of rank $k$ and the new batch-GGN contribution of rank at most $BC$,
> $
> W_t \in \mathbb{R}^{D \times (k+BC)}
> $.
> We then apply truncated SVD and compress the representation back to rank $k$. In our setting, we use $k=C$.
>
> We have clarified this point in the GGN paragraph (page 3).

---

> > ### Comment · Reviewer_2RrQ · 2026-05-21
> > **Thanks for the update**
> >
> > The min(D, BC) point is the one I was looking for, thanks for adding that in. I will update my review.

---

### Author Response · Authors · 2026-05-21

We thank all reviewers for the careful reading and constructive feedback. We have uploaded a revised PDF with the main changes highlighted in red.

At a high level, the revision clarifies the points about the GGN approximation. We also expanded the discussion of dataset size on the costs, computational and memory usage, and scalability, including direct timing and memory comparisons with EWC and OSLA. In addition, we revised the claims around $\mathbf{Q}$ and backward smoothing, clarified the novelty, and added guidance for choosing $\mathbf{Q}$, $k$, curvature batch size, and $\lambda$. We expanded the related-work discussion and clarified the code-release plan.

We appreciate the reviewers' suggestions, which helped us improve both the technical clarity and the positioning of the paper.

---

### Decision · Action_Editor_Wy3N · 2026-06-13

**Recommendation:** Accept as is

**Audience:**

Yes

**Audience Explanation:**

Yes. The paper studies an important problem in continual learning and proposes a novel Bayesian approach with both theoretical and empirical contributions, which should be of interest to a portion of the TMLR audience.

**Claims And Evidence:**

Yes

**Claims Explanation:**

Yes. The main claims are supported by both theoretical analysis and empirical results on several benchmark datasets. The evidence is generally clear, convincing, and consistent with the conclusions drawn in the paper.